# Micro-scale control of oligodendrocyte morphology and myelination by the intellectual disability-linked protein acyltransferase ZDHHC9

Hey-Kyeong Jeong[1†], Estibaliz Gonzalez-Fernandez[1†], Ilan Crawley[1], Julia M Coakley[1], Jinha Hwang[2], Dale DO Martin[1‡], Shernaz X Bamji[3], Jong-Il Kim[2], Shin H Kang[1,4]*, Gareth M Thomas[1,4]*

[1]Center for Neural Development and Repair, Lewis Katz School of Medicine at Temple University, Philadelphia, United States; [2]Department of Biomedical Sciences, Seoul National University College of Medicine, Seoul, Republic of Korea; [3]Department of Cellular and Physiological Sciences, Life Sciences Institute and Djavad Mowafaghian Centre for Brain Health, University of British Columbia, Vancouver, Canada; [4]Department of Neural Sciences, Lewis Katz School of Medicine at Temple University, Philadelphia, United States

*For correspondence:
shin.kang@temple.edu (SHK);
gareth.thomas@temple.edu
(GMT)

[†]These authors contributed
equally to this work

Present address: [‡]Department of
Biology, University of Waterloo,
Waterloo, Canada

Competing interest: The authors
declare that no competing
interests exist.

Reviewing Editor: Klaus-Armin
Nave, Max Planck Institute
for Multidisciplinary Sciences,
Germany

## eLife Assessment

This study provides an in-depth exploration of the impact of X-linked ZDHHC9 gene mutations on cognitive deficits and epilepsy, with a particular focus on the expression and function of ZDHHC9 in myelin-forming oligodendrocytes (OLs). These **valuable** findings offer insights into ZDHHC9-related X-linked intellectual disability (XLID) and shed light on the regulatory mechanisms of palmitoylation in myelination. The experimental design and analysis of results are **solid**, providing a reference for further research in this field.

**Abstract** Mutations in the X-linked *ZDHHC9* gene cause cognitive deficits in humans, with a subset of patients suffering from epilepsy. X-linked intellectual disability (XLID) is often ascribed to neuronal deficits, but here we report that expression of human and mouse ZDHHC9 orthologs is far higher in myelinating oligodendrocytes (OLs) than in other CNS cell types. *ZDHHC9* codes for a protein acyltransferase (PAT), and we found that ZDHHC9 is the most highly expressed PAT in OLs. Wild-type ZDHHC9 localizes to Golgi outposts in OL processes, but other PATs and XLID mutant forms of ZDHHC9 are restricted to OL cell bodies. Using genetic tools for OL progenitor fate tracing and sparse cell labeling, we show that mice lacking *Zdhhc9* have grossly normal OL development but display extensive morphological and structural myelin abnormalities. Consistent with the hypothesis that these deficits are OL-autonomous, they are broadly phenocopied by acute *Zdhhc9* knockdown in cultured conditions. Finally, we found that ZDHHC9 palmitoylates Myelin Basic Protein (MBP) in heterologous cells, and that palmitoylation of MBP is impaired in the *Zdhhc9* knockout brain. Our findings provide critical insights into the mechanisms of *ZDHHC9*-associated XLID and shed new light on the palmitoylation-dependent control of myelination.

# Introduction

Mutations in the *ZDHHC9* gene cause XLID (*Baker et al., 2015*; *Raymond et al., 2007*; *Masurel-Paulet et al., 2014*). XLID-associated *ZDHHC9* mutations include nonsense mutations and missense mutations affecting key amino acids, as well as splice site mutations, insertions, and deletions that result in frameshifts. Importantly, all these genetic changes are predicted to be loss-of-function. These findings suggest that ZDHHC9 is essential for higher brain function.

Given that many XLID genes are neuronally enriched (*Devys et al., 1993*; *Budreck and Scheiffele, 2007*; *Toya et al., 2023*; *Rusconi et al., 2008*; *Guarnieri et al., 2017*; *Murphy et al., 2006*; *Bienvenu et al., 2002*), an a priori hypothesis is that ZDHHC9 acts in neurons. However, patients with *ZDHHC9* mutations have grossly normal gray matter (GM) but reduced brain white matter (WM) volume, especially in the corpus callosum (CC) (*Baker et al., 2015*; *Raymond et al., 2007*; *Masurel-Paulet et al., 2014*). *ZDHHC9* mutations are also linked to cerebral palsy, a condition in which intellectual disability is often associated with WM impairment (WMI) (*Jiang et al., 2019*; *Odding et al., 2006*; *van Eyk et al., 2019*; *McMichael et al., 2015*). These findings raise the possibility that ZDHHC9 may be crucial for normal WM development and/or proper function of oligodendrocytes (OLs), the myelin-forming CNS glia abundant in WM. Notably, *Zdhhc9* knockout (KO) mice exhibit WM volume reductions and seizure-like activity (*Kouskou et al., 2018*; *Shimell et al., 2019*), as well as behavioral phenotypes seen in other mouse models of ID (*Kouskou et al., 2018*). Thus, *Zdhhc9* KO mice have the potential to serve as an excellent model for human ZDHHC9 mutations. However, their phenotype is yet to be fully characterized.

*ZDHHC9* codes for a protein acyltransferase (PAT), an enzyme that catalyzes the modification of protein cysteine residues with palmitate and related fatty acids. This process, termed palmitoylation (also known as *S*-palmitoylation or *S*-acylation), plays a critical role in targeting proteins to specific subcellular membrane locations (*Linder and Deschenes, 2007*). Nearly 30 years ago, it was reported that major myelin proteins in the nervous system, including Proteolipid protein (PLP), Myelin basic protein (MBP), Myelin-associated glycoprotein (MAG), and Myelin oligodendrocyte glycoprotein (MOG), are palmitoylated (*Bizzozero and Good, 1990*; *Agrawal et al., 1983*). However, at that time, methods were lacking to define the roles of palmitoylation of these proteins in myelin formation and function. Later proteomic studies revealed many other palmitoylated myelin proteins (*Kang et al., 2008*; *Wan et al., 2013*; *Gorenberg et al., 2022*; *Edmonds et al., 2017*), and an isolated study suggested that palmitoylation targets myelin proteins to the plasma membrane of cultured OLs (*Schneider et al., 2005*). Despite these findings, little is known regarding how myelin protein palmitoylation is regulated and the functional importance of this process for higher brain function.

We hypothesized that ZDHHC9-dependent palmitoylation plays a critical role in proper myelination and WM formation. Consistent with this notion, we report that ZDHHC9 is the most highly expressed PAT in mouse and human OLs. Moreover, ZDHHC9 localizes to puncta in OL processes that are likely Golgi outposts, whereas other PATs tested and *ZDHHC9* XLID mutant forms are restricted to OL cell bodies. These findings may explain why *ZDHHC9* loss of function cannot be compensated by other PATs. In mice lacking *Zdhhc9*, we did not detect changes in OL lineage cell generation or total MBP expression levels, but we found that at the micro-scale *Zdhhc9* KO OLs are dysmorphic and myelin ultrastructure is highly abnormal, with both hypo- and hypermyelination of axons in WM tracts. We show that with the help of Golga7, a ZDHHC9 partner protein that also localizes to Golgi outposts, ZDHHC9 robustly palmitoylates the major myelin protein MBP, and that MBP palmitoylation is impaired in *Zdhhc9* KO brain. Palmitoyl- and total levels of another myelin protein, Myelin-associated Glycoprotein (MAG) are also affected by *Zdhhc9* KO, suggestive of a broader deficit in myelin protein regulation and organization in these mice. Interestingly, the XLID mutant ZDHHC9-P150S still palmitoylates MBP in a Golga7-dependent manner in transfected non-neuronal cells, raising the possibility that impaired subcellular localization in OLs, rather than impaired catalytic activity, may cause or contribute to WM abnormalities in some cases of *ZDHHC9*-associated XLID. Together, our findings provide new insights into mechanisms of *ZDHHC9*-associated XLID and the palmitoylation-dependent control of myelination, a process first reported decades ago, but about which almost nothing is known.

# Results

## Cell-specific transcriptomics reveals that ZDHHC9 and its partner protein GOLGA7 are enriched in oligodendrocytes in the mouse and human brain

To assess *Zdhhc9* expression in OLs, we performed FACS using cerebral cortex (CTX) of P30 *Mobp-EGFP* BAC transgenic (Tg) mice, which express EGFP only in mature OLs (*Figure 1A*; *Gong et al., 2003*; *Kang et al., 2013*). We isolated RNA from the EGFP⁺ OLs and performed OL-specific RNA sequencing (RNA-Seq). Total RNA was also extracted from the cortices of the same mice without OL sorting and used for a separate bulk RNA-Seq. Bioinformatics analysis indicated that genes previously known to be highly expressed in myelinating OLs (*McMichael et al., 2015*) were highly enriched in our OL-specific RNA preparations (*Figure 1B*). In contrast, expression levels of genes specific to other neural cell types and endothelial cells were very low, confirming that our transcriptomic dataset is highly OL-specific (*Figure 1B*).

We then extracted fragments per kilobase of transcript per million mapped reads (FPKM) values for all ZDHHC family PATs from our OL-specific RNA-Seq dataset. We found that *Zdhhc9* is the most highly expressed PAT in OLs (FPKM ~180), with mRNAs for *Zdhhc14*, *Zdhhc17*, and *Zdhhc20* present at lower but detectable levels (FPKM 30~70) (*Figure 1C*). The remaining PATs were expressed at far lower levels (FPKM <10). These results suggest that ZDHHC9 is the predominant PAT in myelinating OLs.

ZDHHC9 activity requires a conserved partner, Golgi Complex Protein-16 (gene name *Golga7*) (*Shimell et al., 2019*; *Swarthout et al., 2005*; *Nguyen et al., 2023*). We found that *Golga7* was also abundantly expressed in OLs, at far higher levels than other Golga family members, including *Golga7*'s closest paralog, *Golga7b* (*Figure 1D*). FPKM values for *Zdhhc9* and *Golga7* were far lower in total RNA preps obtained from whole cortices without specific cell type isolation (*Figure 1E*, left Heatmap), suggesting that high *Zdhhc9* and *Golga7* expression is an OL-specific feature, rather than a general characteristic of CNS cells. Consistent with this notion, examination of another CNS cell type-specific RNA-Seq dataset, obtained with sorted cells from P17 brain cortices via immunopanning (*Zhang et al., 2014*), confirmed high and specific expression of *Zdhhc9* and *Golga7* in myelin-forming OLs (*Figure 1E*, right Heatmap). Moreover, examination of a human RNA-Seq study (*Zhang et al., 2016*) revealed that, like mouse CNS, both *ZDHHC9* and *GOLGA7* are enriched in human OLs, compared to any other PATs or other GOLGA family members, respectively. (*Figure 1F*). Together, these findings raise the possibility that ZDHHC9, in concert with GOLGA7, plays an important role in OLs.

## Wild-type ZDHHC9 and GOLGA7 localize to oligodendrocyte processes in vitro, unlike other PATs or ZDHHC9 mutants

As a first step toward understanding ZDHHC9 function in OLs, we sought to define the subcellular localization of transfected HA-tagged ZDHHC9 (HA-ZDHHC9) in cultured OLs. We also compared the subcellular localization of HA-ZDHHC9 with that of HA-tagged forms of ZDHHC3, ZDHHC7, and ZDHHC17, three other ZDHHC-PATs that are implicated in nervous system regulation and which are expressed in OLs in vivo (*Figure 1E*, *Zhang et al., 2014*; *Zeisel et al., 2018*; *Fang et al., 2006*; *Noritake et al., 2009*; *Sanders et al., 2016*; *Niu et al., 2020*; *Kerkenberg et al., 2021*).

After three days of culture in differentiation medium (*Figure 2A*), cultured OLs were extensively ramified and showed strong MBP immunofluorescence ('OL3D'; *Figure 2—figure supplement 1A*). We transfected these cultured OLs and fixed them 9 hr later to minimize the likelihood of ZDHHC-PAT expression itself altering OL morphology and affecting HA-PAT subcellular localization (*Figure 2A*). Under these conditions, HA-ZDHHC3, HA-ZDHHC7, and HA-ZDHHC17 were only detected in OL cell bodies, consistent with their restriction to the somatic Golgi in other cell types (*Fang et al., 2006*; *Niu et al., 2020*; *Collura et al., 2020*; *Ohno et al., 2006*; *Greaves et al., 2008*; *Figure 2B and D*). In contrast, HA-ZDHHC9 was detected both in OL cell bodies and in discrete puncta in OL processes (*Figure 2B and D*). Within OL cell bodies, ZDHHC9 partially colocalized with the Golgi marker GM130 (*Figure 2—figure supplement 1B*). Consistent with this Golgi localization, ZDHHC9 also colocalized in part with additional Golgi markers (TGN38, Giantin; *Figure 2—figure supplement 1C and D*). However, these markers were not detected in OL processes and therefore did not colocalize with ZDHHC9 in this latter location (*Figure 2—figure supplement 1A–D*).

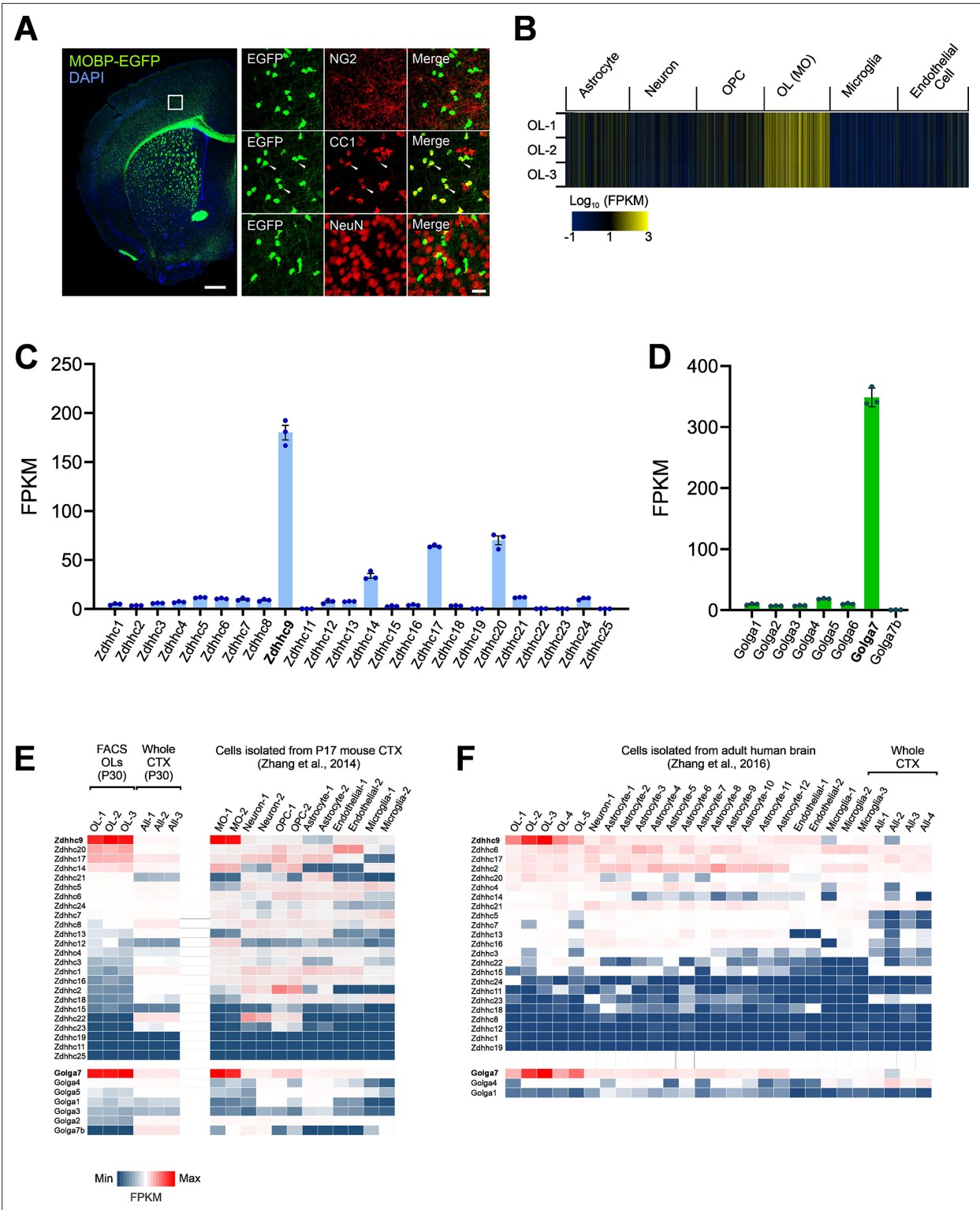

**Figure 1.** Oligodendrocyte-specific transcriptomics reveals oligodendrocyte (OL)-enriched expression of ZDHHC9 and GOLGA7. (**A**) Fluorescent image of widespread EGFP expression in the brain of *Mobp-EGFP* mice (Left). Confocal images of EGFP and other cell-specific markers (NG2 for OPC, CC1 for OLs, and NeuN for neurons) (Right). White arrowheads indicate colocalization of EGFP+ cells and CC1-immunoreactivities. Boxed area in left panel corresponds to the region used to acquire confocal images for NG2 and CC1 in right panel. Scale bars: 500 (left) and 20 (right) μm. (**B**) Heatmap for expression levels of previously reported cell type-specific gene clusters from the OL-specific RNA-Seq results. EGFP+ OLs were isolated from cortices of three *Mobp-EGFP* mice with fluorescence-activated cell sorting (FACS), and their RNAs were used for RNA-seq. (**C, D**) Fragments per kilobase per

*Figure 1 continued on next page*

*Figure 1 continued*

million (FPKM) values of ZDHHC-protein acyltransferases (PATs) (**C**) and Golgin subfamily A members (**D**) expressed in OLs. Both graphs show mean $\pm$ SEM. (**E**) Heatmap of relative expression of all protein acyltransferase (PATs) and Golgin subfamily A member genes in mouse OLs and other brain cells according to data from this study (left) and *Zhang et al., 2014* (right). (**F**) Heatmap of relative expression of all PATs and Golgin subfamily A member genes in human OLs and other brain cells according to *Zhang et al., 2016*.

We also compared the subcellular distribution of wild-type ZDHHC9 (ZDHHC9WT) with that of XLID-associated ZDHHC9 mutants (R96W; R148W; P150S) (*Han et al., 2017*; *Schirwani et al., 2018*). In contrast to HA-ZDHHC9WT, these XLID-associated HA-ZDHHC9 mutants did not localize to OL processes (*Figure 2C and D*). These results raise the possibility that dysregulated subcellular localization of ZDHHC9, rather than, or in addition to, reduced catalytic activity (*Mitchell et al., 2014*) is a causative factor in *ZDHHC9* loss of function mutations in XLID.

We also sought to define the localization in OLs of Golga7, which directly binds and enhances function of ZDHHC9 in other cell types (*Shimell et al., 2019*; *Swarthout et al., 2005*; *Nguyen et al., 2023*). Consistent with this role, myc-tagged Golga7 (myc-Golga7) colocalized extensively with HA-ZDHHC9 in both OL cell bodies and in discrete puncta in OL processes (*Figure 2—figure supplement 2A*). We then sought to further define the identity of these discrete ZDHHC9/Golga7-positive puncta. In neurons, specialized Golgi outposts and Golgi satellites can function as acentrosomal microtubule-organizing centers (MTOCs) in dendrites (*Ori-McKenney et al., 2012*; *Valenzuela and Perez, 2015*; *Mikhaylova et al., 2016*; *Wang and Gleeson, 2020*; *Govind et al., 2021*). Interestingly, Golgi outposts were also recently reported to be present in OLs (*Fu et al., 2019*), and a subset of OL Golgi outposts can be marked by the proteins TPPP and Mannosidase II (ManII) (*Mikhaylova et al., 2016*; *Govind et al., 2021*; *Fu et al., 2019*). Consistent with their assignation as Golgi outposts, ZDHHC9-positive puncta in OL processes colocalized with GFP-tagged ManII (ManII-GFP) and, to a lesser extent, with Flag-tagged TPPP (TPPP-Flag) (*Figure 2—figure supplement 2B and C*).

The OL processes in our culture condition, most of which are MBP$^+$ even at this comparatively short time point post-differentiation (*Figure 2—figure supplement 1*) go on to develop into large lipid-rich membranous sheets (*Figure 2—figure supplement 1A*). In vivo, these OL processes form a spiral membrane expansion on axons (i.e. myelination) in vivo (*Fitzner et al., 2006*). ZDHHC9's localization to Golgi outposts/satellites in processes of cultured OLs suggest its potential role in myelin formation (myelination) and/or maintenance. Moreover, this function of ZDHHC9 might not be shared with other PATs tested and may be impaired in XLID-associated mutant forms of ZDHHC9.

## No detectable changes in oligodendrocyte development or gross myelination in *Zdhhc9* KO mice

To address whether ZDHHC9 regulates OL development and myelination in vivo, we examined the brains of *Zdhhc9* KO mice. Histological observations of brain sections from 6-week-old male mice revealed no apparent difference in MBP-labeled OL processes between WT control and *Zdhhc9* KO mice (*Figure 3A*). OL numbers in CTX and the corpus callosum (CC), assessed using OL markers aspartoacylase (ASPA) or Quaking 7 (recognized by antibody CC1), also did not detect any significant difference between the two groups (*Figure 3—figure supplement 1A–C*). In addition, preliminary analysis of the early OL marker breast carcinoma amplified sequence 1 (BCAS1) did not reveal obvious differences between the two genotypes (*Figure 3—figure supplement 1D*). To quantify OLs more precisely, we crossed *Zdhhc9* KO with *Mobp-EGFP* mice and counted EGFP-labeled OLs. Again, we did not detect significant differences in EGFP$^+$ OL density between the two genotypes in three examined CNS regions (CTX, CC, and spinal cord), at two different ages (P28, P56; *Figure 3B and C*). We also detected no difference in the density of NG2$^+$ OL progenitor cells (OPCs) between WT and *Zdhhc9* KO mice (*Figure 3D and E*).

We next asked whether *Zdhhc9* loss affects the rate of oligodendrogenesis (maturation of OPCs to OLs). To address this question, we crossed *Zdhhc9* KO mice with *Pdgfra-CreER; Rosa26-lsl-EGFP (RCE)* mice (*Sousa et al., 2009*) and analyzed the fates of genetically labeled OPCs. Control (*Pdgfra-CreER; RCE*) and KO (*Pdgfra-CreER; RCE; Zdhhc9$^{y/-}$ or $^{-/-}$*) mice received tamoxifen injections at P21 and were sampled 3 weeks later (P21 +21) (*Figure 3F-1*). This allowed us to label PDGFRα$^+$ OPCs with EGFP at P21 and track their differentiation (*DeFlitch et al., 2022*). Newly differentiated EGFP$^+$ OLs from the previously labeled EGFP-labeled OPCs were analyzed (*Figure 3F-2, G*). However, we did

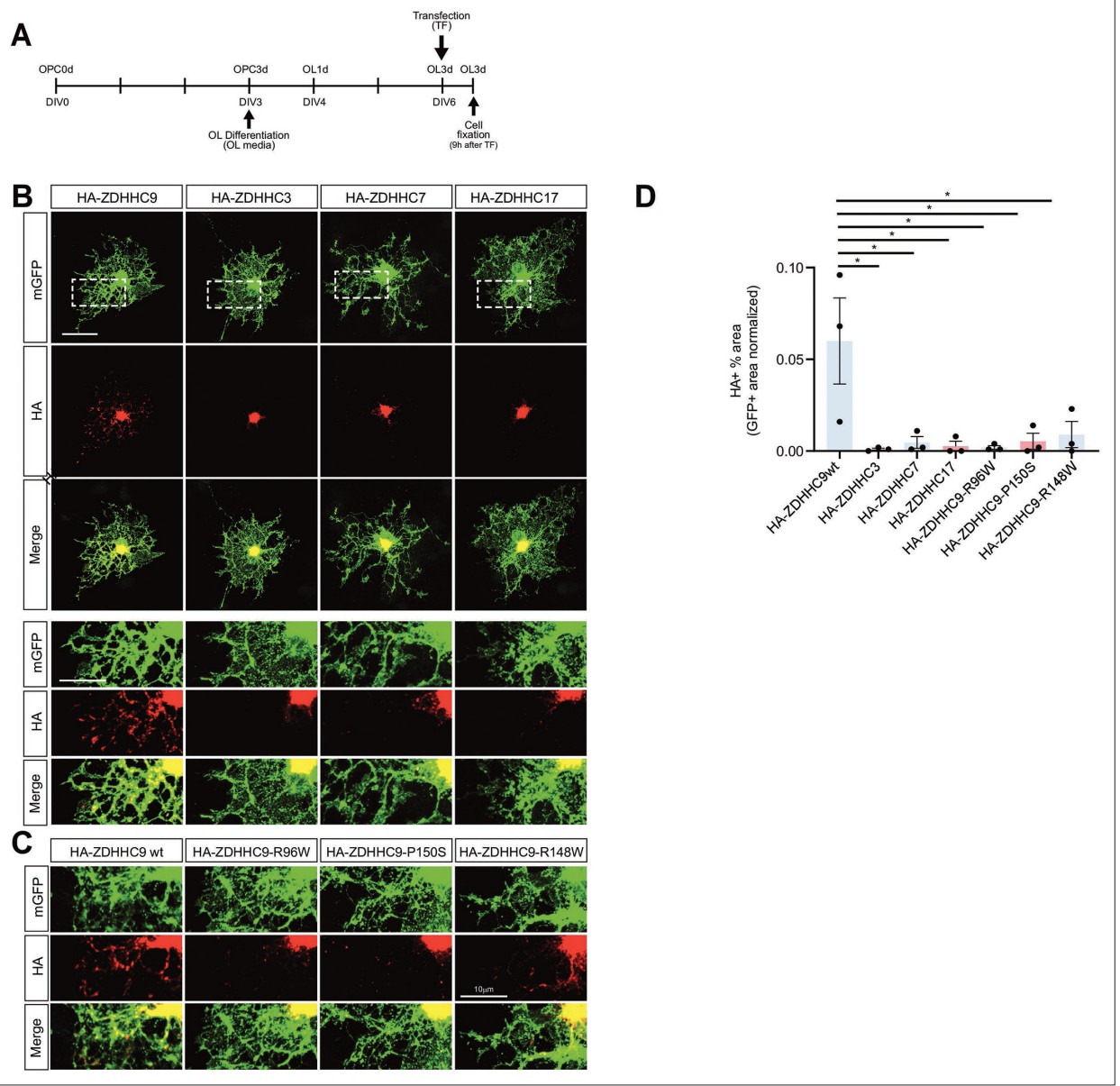

**Figure 2.** HA-ZDHHC9WT localizes to puncta in oligodendrocyte (OL) processes, but other protein acyltransferases (PATs), and X-linked intellectual disability (XLID) mutant forms of ZDHHC9, are restricted to OL cell bodies. (**A**) Experimental timeline. (**B**) Images of morphologically mature OLs transfected as in *A* to express the indicated HA-tagged PATs and immunostained with the indicated antibodies. Scale bar: 20 μm. Lower panels show enlarged images of the boxed regions in upper panels. Scale bar: 10 μm. (**C**) Images as in lower panels of *B* of OLs transfected to express WT HA-ZDHHC9 or the indicated XLID mutant forms of ZDHHC9. (**D**) Quantified data confirms that HA-ZDHHC9 WT occupies a greater percentage of the total OL area (GFP +area) compared with other PATs, or with HA-ZDHHC9 XLID mutants. *p=0.032 (HA-ZDHHC9WT vs HA-ZDHHC3); p=0.0053 (HA-ZDHHC9WT vs HA-ZDHHC7); p=0.004 (HA-ZDHHC9WT vs HA-ZDHHC17); p=0.0036 (HA-ZDHHC9WT vs HA-ZDHHC9-R96W); p=0.0058 (HA-ZDHHC9WT vs HA-ZDHHC9-P150S); p*P*=0.0047 (HA-ZDHHC9WT vs HA-ZDHHC9-R148W), Data are from 3 to 5 cells per condition, pooled from n=3 cultures. Data are plotted as mean ± SEM.

The online version of this article includes the following figure supplement(s) for figure 2:

**Figure supplement 1.** In oligodendrocyte (OL) cell bodies, ZDHHC9 localizes to somatic Golgi.

**Figure supplement 2.** ZDHHC9-positive puncta in oligodendrocyte (OL) processes are Golgi outposts/satellites.

not detect differences in the number of newly formed EGFP-labeled ASPA⁺ OLs and % of EGFP⁺ OLs (*Figure 3H, I*) and the ratio of OLs to OPCs among EGFP-labeled cells (data not shown) between the two groups. These results suggest that *Zdhhc9* KO does not alter oligodendrogenesis in either the young or mature CNS, at least using the markers and methodologies that we employed.

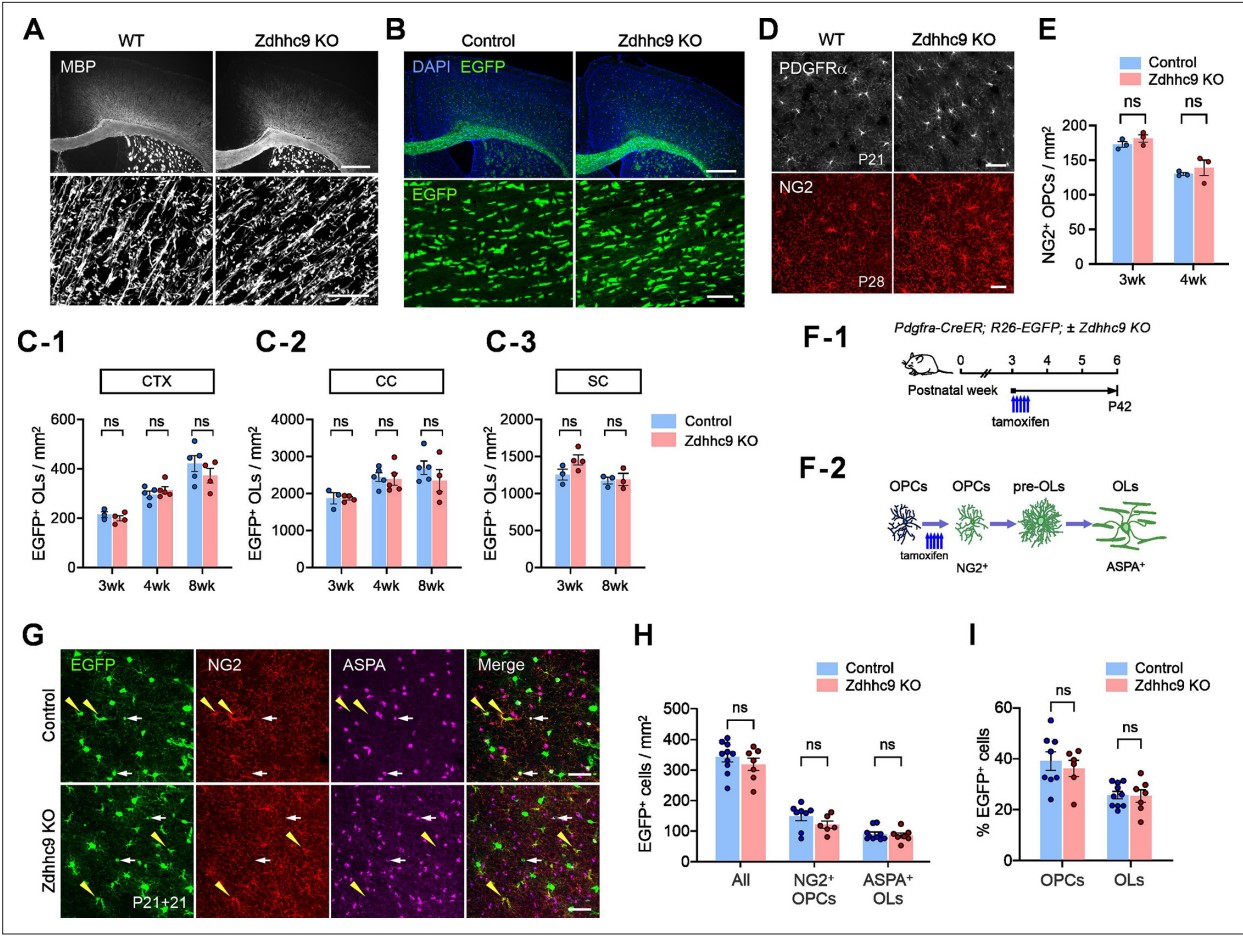

**Figure 3.** No detectable gross abnormality in oligodendrocyte development in *Zdhhc9* KO mice. (**A**) Fluorescent (upper) and confocal (bottom) images of myelin basic protein (MBP) immunostaining in the brain of 6-week-old WT and *Zdhhc9* KO mice. MBP confocal images were taken from layers IV/V of CTX. Scale bars: 500 (upper panel) and 20 (bottom) μm. (**B**) Fluorescent (upper) and confocal (bottom) images of EGFP in the brain of 8-week-old control (*Mobp-EGFP*) and *Zdhhc9* KO (*Mobp-EGFP; ZDHHC9⁻/⁻*) male mice. EGFP confocal images were taken from the CC. Scale bars: 500 (upper panel) and 50 (bottom) μm. (**C1-C3**) Quantification of EGFP⁺ OLs in CTX, CC, and spinal cord (SC) from 3 week (P21), 4 week (P28), and 8 week (P56)-old control and *Zdhhc9* KO *Mobp-EGFP* mice (n=3–5 mice for each group). (**D**) Confocal images of cortical PDGFRα⁺ OPCs from 3-week-old male mice (upper panel) and NG2⁺ OPCs from 8-week-old male mice (lower panel). Scale bar: 50 μm. (**E**) Quantification of the density of NG2⁺ OPCs in CTX of 3 week and 4-week-old WT and Zdhhc9 KO mice (n=3 per group). (**F-1**) Experimental scheme for OPC fate tracing. *Pdgfra-CreER; RCE* and *Pdgfra-CreER; RCE; Zdhhc9* KO mice received tamoxifen injections starting at P21 for a duration of 3 days, and their brains were collected at P42. (**F-2**) Schematic diagram illustrating the stepwise changes in genetically labeled OPCs during fate tracing. (**G**) Confocal images of EGFP, NG2, and ASPA in CTX of control and *Zdhhc9* KO *Pdgfra-CreER; RCE* mice (P21 + 21). Yellow arrowheads: overlapping signals between EGFP and NG2; white arrows: overlapped signals between EGFP and ASPA. Scale bars: 50 μm. (**H**) Quantification of EGFP⁺NG2⁺ OPCs and EGFP⁺ASPA⁺ OLs. (**I**) Percentage of NG2⁺ OPCs and ASPA⁺ OLs among EGFP⁺ cells. Data are mean ± SEM (**E, H, I**). For *H* and *I*, N=10 (control) or 7 (Zdhhc9 KO) male and female mice. Two-way ANOVA and subsequent pair-wise comparisons were performed with Šidák's multiple comparison tests for (**C, E, H, I**). ns: not significant.

The online version of this article includes the following figure supplement(s) for figure 3:

**Figure supplement 1.** No alteration of oligodendrocyte density in *Zdhhc9* KO mice.

## *Zdhhc9* KO impairs microstructure of oligodendrocyte processes

Although we did not detect differences in overall oligodendrogenesis or myelin production, we reasoned that loss of *Zdhhc9* might still affect the targeting of specific myelin proteins to the membrane, resulting in irregular formation of processes in OLs or abnormalities in myelin. To address this possibility, we used a sparse genetic OL labeling method, crossing *Mobp-iCreER; Rosa26-mtdTomato-mEGFP (mT/mG)* mice to *Zdhhc9* KO mice. By P56, a small degree of tamoxifen-independent (leaky) Cre activity leads to spontaneous expression of membrane-anchored EGFP (mEGFP) in a subset (~5%) of cortical OLs in these mice. The sparsely EGFP-labeled brain sections were then imaged with confocal microscopy to detect mEGFP signal in CTX and to subsequently trace OL processes (**Figure 4A**). The

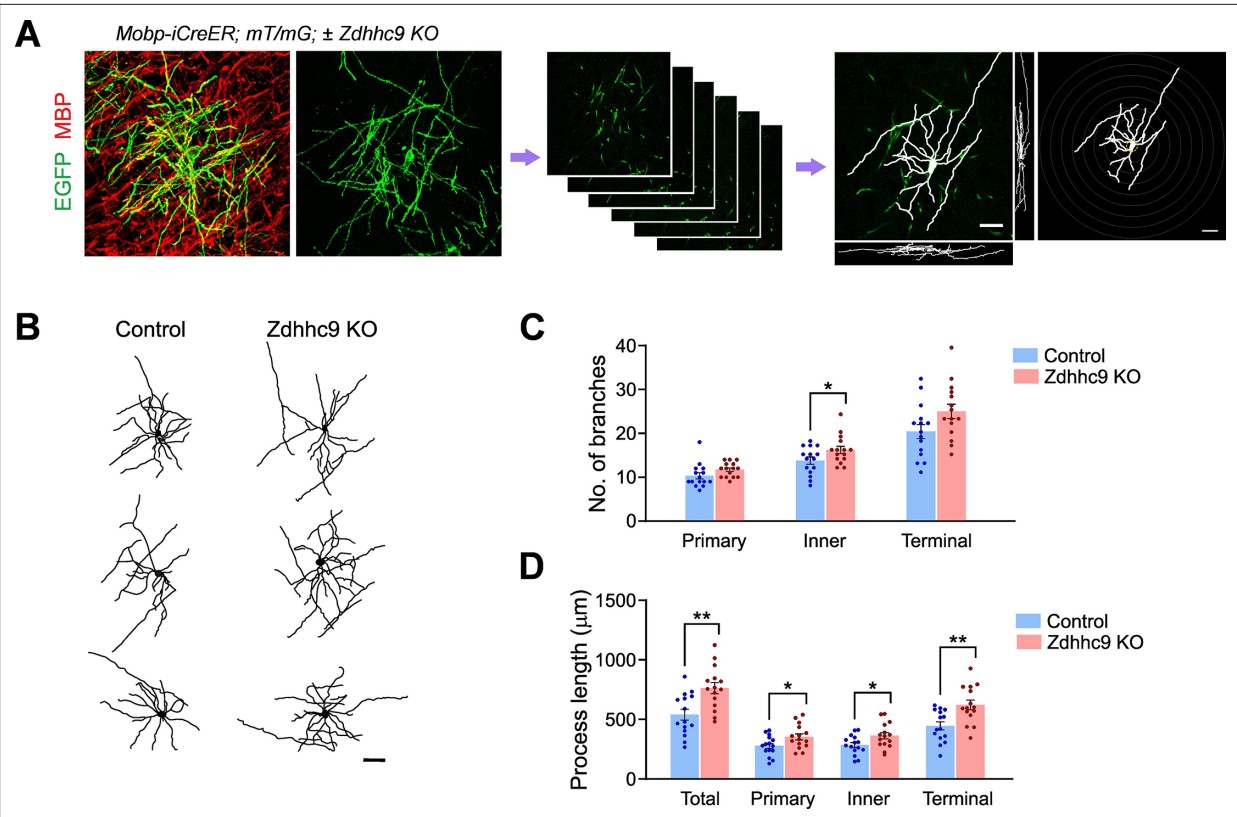

**Figure 4.** Genetic sparse cell labeling reveals altered complexity of oligodendrocyte processes in *Zdhhc9* KO mice. (**A**) Experimental flow of the sparse genetic labeling of oligodendrocytes (OLs), from acquisition of confocal images of mEGFP from cerebral cortex (CTX) in P56 *Mobp-iCreER; mT/mG* mice to 3D OL process tracing to Sholl analysis. mEGFP signal allows for detailed process morphology of individual OLs, compared to more complex 'bulk' MBP signal. (**B**) Representative results from tracing of OL processes. Scale bars: 20 μm. (**C, D**) Branch numbers (**C**) and process length (**D**) were compared between control (*Mobp-iCreER; mT/mG*) and *Zdhhc9* KO (*Mobp-iCreER; mT/mG; Zdhhc9* KO) mice. N=15 OLs (five OLs per mouse, three mice per group). Two-way ANOVA and pair-wise comparisons were performed with Šidák's multiple comparison tests (**C and D**). ns: not significant. *p<0.05; **p<0.01. Data are mean ± SEM.

morphology of individual OLs was then reconstructed as a 3D skeleton (*Figure 4B*). Sholl analysis of these reconstructions revealed that the overall OL process complexity was slightly increased in *Zdhhc9 KO* mice (*Figure 4C*). More surprisingly, the primary and secondary branch process were longer in *Zdhhc9* KO than control mice (*Figure 4D*).

We also analyzed the processes of individual OLs in raw images from these sections. In control mice, the EGFP-labeled processes originating from one OL soma were of similar thickness, with a uniformly smooth structure. However, the thickness of mEGFP+ processes in *Zdhhc9* KO mice was far more heterogeneous, with several spheroid-like swellings (*Figure 5A and B*). The abnormal thickness of EGFP+ OL processes may reflect dysregulated axon recognition by OLs. Notably, the abnormal spheroid-like swellings on OL processes were distinguished from OL somas as they lack signal for DAPI or Olig2 (*Figure 5C*, *Figure 5—figure supplement 1A, B*). Careful tracing of EGFP+ OL processes connected to DAPI+ cell body and quantification of spheroid-like swellings devoid of DAPI signal (*Figure 5C-1, C-2*) revealed significant structural abnormalities in OL processes in *Zdhhc9* KO mice (*Figure 5D and E*). These results indicate that, in contrast to its lack of effect at the gross level, *Zdhhc9* loss greatly alters the structure of individual OL processes at the microscopic level, presumably due to abnormal OL membrane expansion.

## Impaired myelination in Zdhhc9 KO Mice

The non-uniformity of *Zdhhc9* KO OL processes suggests that *Zdhhc9* loss may disrupt the typical bias of myelination based on axon caliber and/or the spreading of the OL membrane along the axon. To address this issue, we examined the extent and pattern of axonal myelination using electron

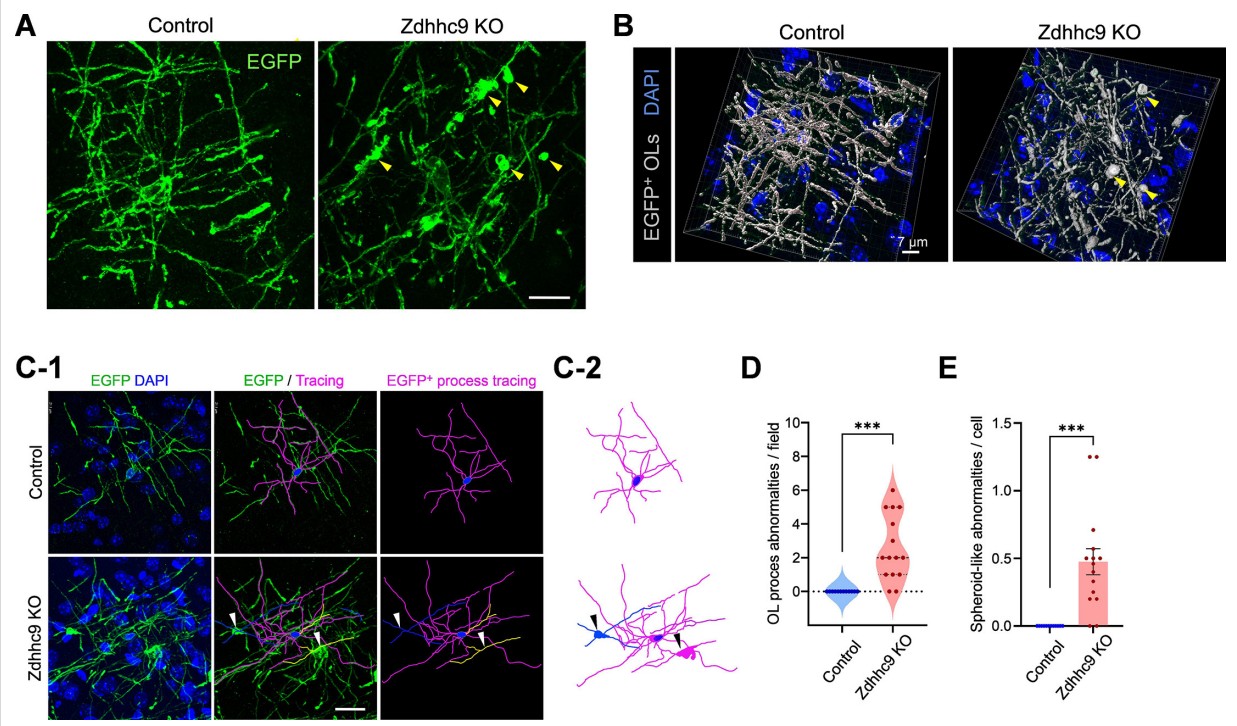

**Figure 5.** Abnormal oligodendrocyte processes in *Zdhhc9* KO mice. (**A**) Confocal microscopy of mEGFP in P56 *Mobp-iCreER; mT/mG* mice reveals morphological abnormalities, such as non-homogenous EGFP⁺ cell processes and spheroid-like membrane folding (yellow arrowheads). Scale bar: 20 μm. (**B**) 3D-reconstruction of EGFP⁺ oligodendrocytes (OLs) and DAPI⁺ nuclei from the images shown in (**A**) with the Imaris software. Scale bar: 7 μm. (**C**) OL processes and connected OL cell body were traced. Two different cells are shown in magenta and green. Arrowheads indicate spheroid-like swelling. (**D, E**) Quantification of process abnormalities per field (**D**) and per cell (**E**). N=15 cells from 3 mice per group. Student's t-test. ***p<0.001.

The online version of this article includes the following figure supplement(s) for figure 5:

**Figure supplement 1.** Spheroid-like abnormal structures in *Zdhhc9* KO mice are distinct from Olig2⁺ OL cell bodies.

microscopy (EM) (*Figure 6A*). In electron micrographs from P56 CC, the number of axons did not differ between WT and *Zdhhc9* KO mice (*Figure 6A and B*). However, while most axons were evenly myelinated in WT mice, myelin patterns of axons in *Zdhhc9* KO mice were highly abnormal; with many large axons unmyelinated (*Figure 6A and C*) and a subset of small diameter axons (<0.5 μm) appearing to be hypermyelinated (*Figure 6A and D*). Consistent with the latter finding, *g*-ratios of these small diameter axons were smaller in *Zdhhc9* KO mice (*Figure 6E and F*).

Next, we inquired whether the dysmyelination observed in young adult (P56) *Zdhhc9* KO mice resulted from impaired initial myelination or from impaired myelin maintenance or active demyelination. At P30, a time at which myelination is actively ongoing, EM images showed a broadly similar extent of myelination across all axons in WT mice (*Figure 6G and H*). However, in *Zdhhc9* KO mice, both hypo- and hypermyelination of axons was already noticeable at this developmental stage (*Figure 6G*), which was evident from an increased interquartile range of *g*-ratios (*Figure 6H, I*). These findings suggest that *Zdhhc9* loss results in dysmyelination and, thus, that ZDHHC9 is required for proper initial myelination of individual axons.

## Evidence supporting a cell-autonomous role of ZDHHC9 in Oligodendrocyte Maturation

*Zdhhc9* is constitutively deleted in the KO mice, so OL morphological abnormalities and myelination deficits seen in this line might be caused by loss of action of ZDHHC9 in other cell types. As a first step to testing whether phenotypes in *Zdhhc9* KO mice are cell-autonomous, we delivered lentivirus expressing GFP plus a specific shRNA (*Shimell et al., 2019*) to knock down *Zdhhc9* in WT OPCs in culture and induced OPC-to-OL differentiation one day later (*Figure 7A*). *Zdhhc9* knockdown OLs had

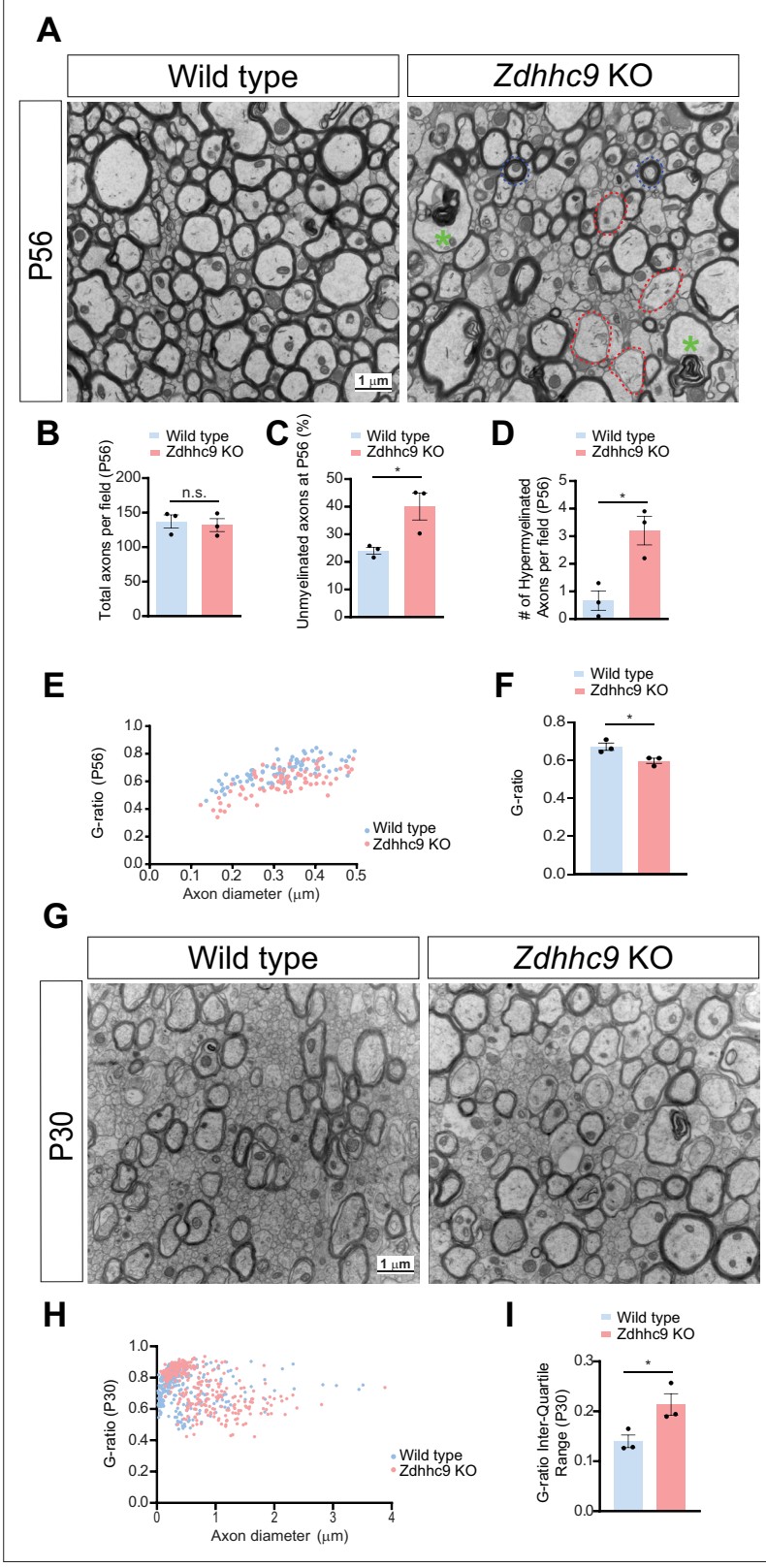

**Figure 6.** Altered myelination in *Zdhhc9* KO corpus callosum in vivo. (**A**) Electron micrographs of corpus callosum axons from P56 male mice of the indicated genotypes. Wild-type axons are uniformly myelinated, but in *Zdhhc9* KO, some large axons are hypomyelinated (red dotted outline), while a subset of small axons appear hypermyelinated (blue dotted outline). There is also frequent dysmyelination (green asterisk). (**B**) Quantification

*Figure 6 continued on next page*

*Figure 6 continued*

of images from **A** reveal no change in total number of axons in *Zdhhc9* KO. n.s.: not significant (p=0.7117, t test, N=3 mice per genotype). (**C**) Percentage of unmyelinated axons from mice of the indicated genotypes (N=3 mice per genotype; *p=0.0036, t-test). (**D**) Number of hypermyelinated small axons per field in corpus callosum images from mice of the indicated genotypes (N=3 mice per genotype; *p=0.0150, t test). (**E**) G-ratio of small diameter axons (0.1–0.5 µm) from mice of the indicated genotypes. (**F**) Average G-ratio from **E** confirms hypermyelination of small diameter axons in P50 *Zdhhc9* KO mice. (**G**) as **A**, but from P30 mice. Deficits in myelination are already evident at this time. (**H**) As **E**, but for all axons from **G**. (**I**) Increased heterogeneity of myelination in *Zdhhc9* KO corpus callosum at P30, represented by an increased interquartile range of G-ratios (*p=0.0425, t-test). Data in **B, C, D, F, I** are mean ± SEM.

significantly reduced expression of MBP, compared to OLs infected to express a scrambled shRNA (*Figure 7B–D*).

We asked whether this failure to express and distribute MBP is due to impaired commitment to the OL lineage in culture. However, *Zdhhc9* knockdown OLs still expressed 2',3'-Cyclic nucleotide 3'-phosphodiesterase (CNP), an early myelin-specific protein of developing OLs (*Baumann and Pham-Dinh, 2001*; *Figure 7—figure supplement 1*). This result suggests that ZDHHC9 is not required for initial differentiation of OPCs to OLs, consistent with the normal number of OLs seen in *Zdhhc9* KO mice in vivo (*Figure 3C*, *Figure 3—figure supplement 1C*). However, consistent with the reduced and restricted MBP expression, *Zdhhc9* knockdown OLs were also morphologically immature (*Figure 7E*) with a reduced degree of branching confirmed by Sholl analysis (*Figure 7F*). These data indicate that *Zdhhc9* loss cell-autonomously impairs OL maturation in vitro.

## A ZDHHC9-Golga7 PAT complex directly palmitoylates MBP

Finally, we sought to identify potential ZDHHC9 substrates in OLs. Myelination of CNS axons by OLs requires the targeting of myelin proteins, including MBP, PLP, MOG, and MAG, to the myelin membrane. These myelin proteins are all palmitoylated (*Kang et al., 2008*; *Schneider et al., 2005*; *Greer et al., 2001*; *Pinner et al., 2016*; *Blanc et al., 2015*), but the PAT(s) that controls myelin protein palmitoylation was not previously identified. We thus used a non-radioactive palmitoylation assay, acyl biotin exchange (ABE) (*Wan et al., 2007*; *Thomas et al., 2012*), to determine whether ZDHHC9 can directly palmitoylate MBP in co-transfected HEK293T cells. Palmitoylation of 21.5 kDa MBP (one of two MBP isoforms that contains a cysteine residue and is thus capable of undergoing palmitoylation) was very low when transfected alone. MBP palmitoylation remained very low when either HA-ZDHHC9 or myc-Golga7 were co-transfected in isolation (*Figure 8A and B*). However, MBP palmitoylation was greatly increased by co-transfection of HA-ZDHHC9 and myc-Golga7 (*Figure 8A and B*).

To examine whether ZDHHC9 palmitoylates MBP in vivo, we used ABE to purify palmitoyl-proteins from forebrain WM tissue (CC and striatum) of WT and *Zdhhc9* KO mouse brains. Consistent with in vivo immunostaining results (*Figure 3A*), *Zdhhc9* KO did not affect total MBP levels measured by western blotting (all isoforms assessed together; *Figure 8C and D*). Two MBP bands were detected in ABE (palmitoyl) fractions, which based on their molecular weights, likely represent the 17.0 and 21.5 kDa isoforms of MBP (*Akiyama et al., 2002*). Like 21.5 kDa MBP, the 17.0 kDa form of MBP contains a cysteine residue and may be subject to palmitoylation (*Akiyama et al., 2002*). Importantly, palmitoylation, but not total expression, of both the 17 kDa and 21.5 kDa MBP isoforms was significantly reduced in *Zdhhc9* KO mice (*Figure 8C and D*). Total and palmitoyl- levels of MAG were also significantly reduced in *Zdhhc9* KO mice, although the palmitoyl:total ratio of MAG was not (*Figure 8—figure supplement 1A and B*). In the same samples, neither palmitoyl-, total, nor palmitoyl:total levels of Cadm4 were affected in *Zdhhc9* KO mice (*Figure 8—figure supplement 1C and D*). This latter finding is consistent with a report ascribing Cadm4 palmitoylation to a different PAT (*Chang et al., 2022*). Together, these results suggest that ZDHHC9 directly palmitoylates MBP with the support of Golga7 and that *Zdhhc9* loss impairs MBP palmitoylation. In addition, *Zdhhc9* loss impacts other myelin protein levels and/or regulation.

Finally, we compared the ability of ZDHHC9wt and ZDHHC9 XLID mutants to palmitoylate MBP. We found that MBP palmitoylation by the XLID mutant ZDHHC9-R148W is almost undetectable in cotransfected heterologous cells and was minimally increased by additional cotransfection of Golga7. Surprisingly, though, ZDHHC9-R96W palmitoylated MBP only approximately 50% as

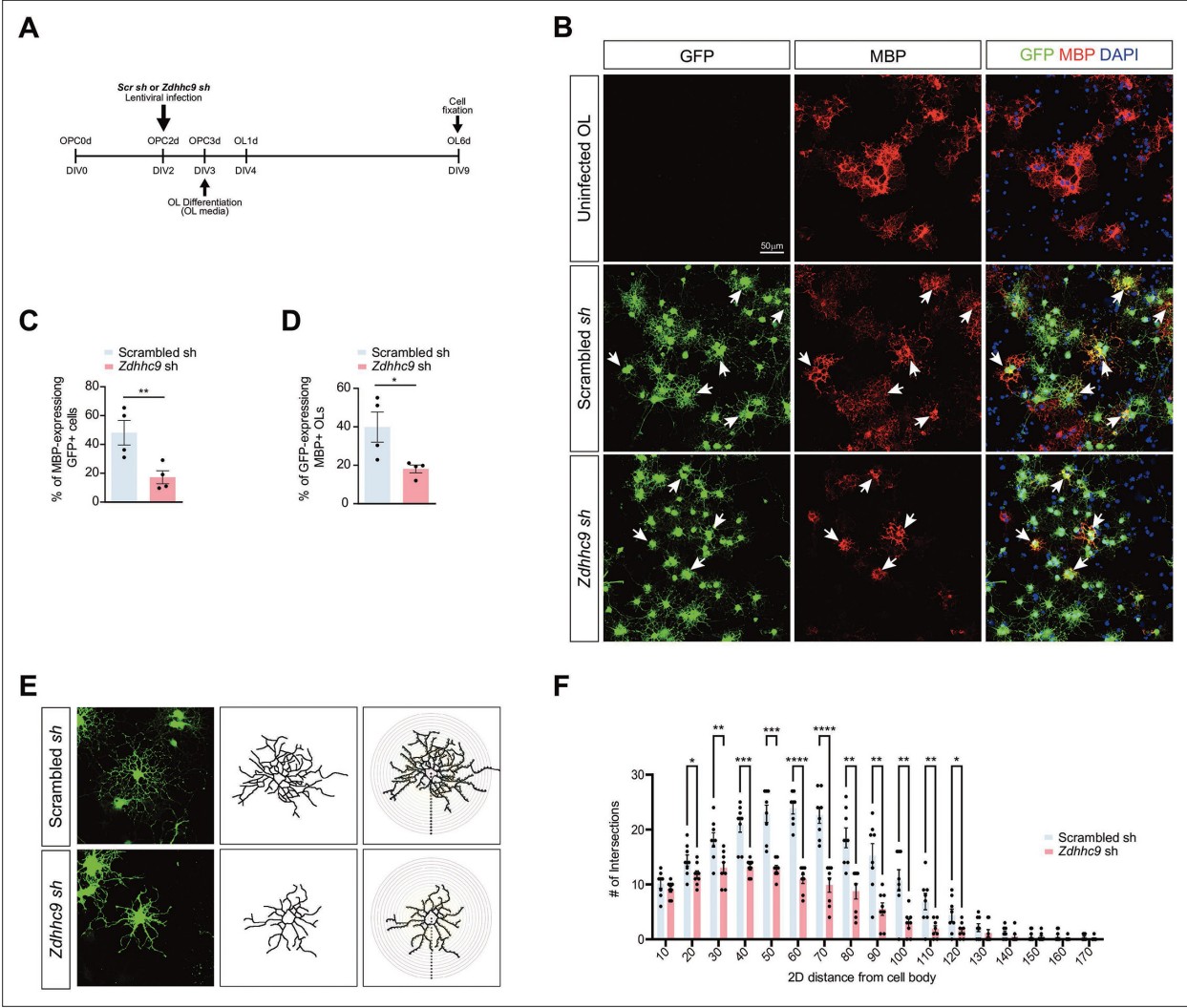

**Figure 7.** ZDHHC9 loss cell-autonomously impairs maturation of cultured oligodendrocytes (OLs). (**A**) Timeline of experiment. (**B**) Immunofluorescent images of cultured OLs, after infection with the indicated lentiviruses and fixation as in *A*. (**C**) Quantified data from *B* reveal that *Zdhhc9* knockdown reduces the percentage of GFP-expressing (virally infected) MBP$^+$ cells i.e., mature OLs (**p=0.0067, t-test, n=4 individual cultures per condition). (**D**) Likewise, *Zdhhc9* knockdown reduces the percentage of myelin basic protein (MBP)-expressing GFP + cells. (*p=0.0359, t-test, n=4 individual cultures per condition). (**E**) Left: Images of individual OLs after infection and fixation as in *B*. *Middle column:* reconstructed outline of individual OLs. *Right column:* Images from middle column with superimposed concentric circles for Sholl analysis. (**F**) Sholl analysis from OLs reconstructed as in *E* confirms reduced morphological complexity of *Zdhhc9* knockdown OLs. n=8 cells per condition. *p<0.05; **p<0.01; ***p<0.001; ****p<0.0001, individual t-tests. Data are mean ± SEM.

The online version of this article includes the following figure supplement(s) for figure 7:

**Figure supplement 1.** Zdhhc9 knockdown causes morphological immaturity of committed OLs.

effectively as ZDHHC9wt in the presence of Golga7. Moreover, ZDHHC9-P150S did not differ significantly from ZDHHC9wt in its ability to palmitoylate MBP in the presence of Golga7. These findings suggest that certain ZDHHC9 XLID mutants are still capable of palmitoylating substrates, at least in heterologous cells. Moreover, autopalmitoylation (sometimes used as a surrogate marker of ZDHHC-PAT activity) of ZDHHC9wt and ZDHHC9-P150S also did not differ significantly in the presence of Golga7, suggesting that ZDHHC9-P150S may have sufficient catalytic activity to drive palmitoylation of an array of substrates. These findings further suggest that dysregulated subcellular localization may be a more important factor than altered catalytic activity in certain cases of *ZDHHC9*-associated XLID.

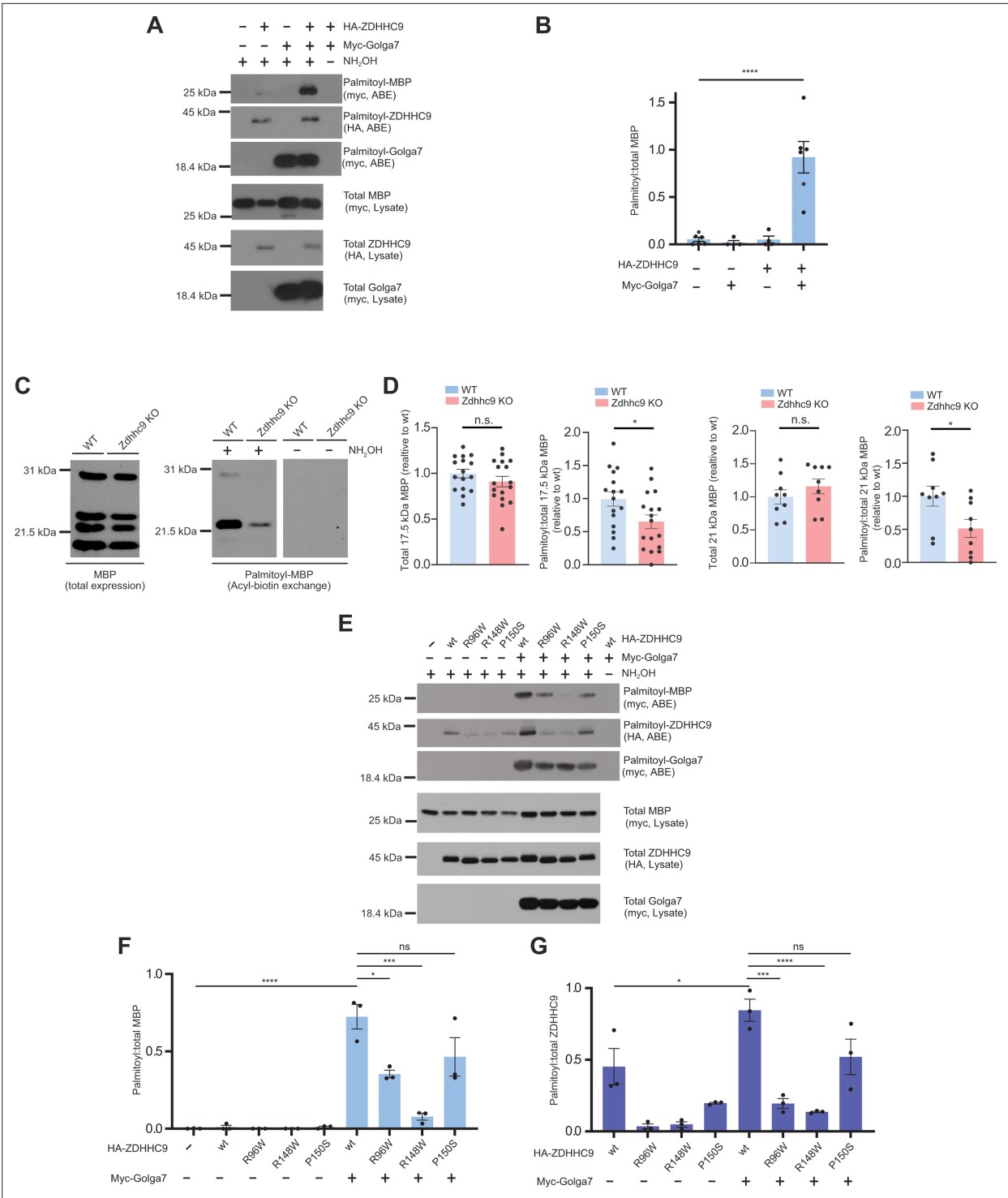

**Figure 8.** ZDHHC9 palmitoylates myelin basic protein (MBP) in cultured cells and in vivo, and certain ZDHHC9 X-linked intellectual disability (XLID) mutants display residual protein acyltransferase (PAT) activity towards MBP. (**A**) Western blots of palmitoyl- and total proteins from HEK293T cells transfected to express the indicated cDNAs. (**B**) Quantified data from C confirms that co-transfection of HA-ZDHHC9 and myc-Golga7 greatly increases MBP palmitoylation. No signal for palmitoyl-MBP is seen in the absence of the key reagent hydroxylamine ($NH_2OH$), confirming assay specificity. ****$p<0.0001$, ANOVA with Dunnett's multiple comparison test. (**C**) Western blots to detect MBP in total lysates and acyl biotin exchange (ABE) fractions from forebrain white matter (WM) (corpus callosum, CC and striatum) from mice of the indicated genotype. (**D**) Quantified data from *C* confirms that *Zdhhc9* loss reduces palmitoyl, but not total, levels of 17 kDa and 21.5 kDa MBP isoforms. palmitoyl:total 17.5 kDa isoform, n=15–16 per genotype; **$p=0.0078$, palmitoyl:total 21 kDa isoform, n=8 per genotype,. 'N' number is lower for 21.5 kDa MBP because in a subset of experiments this isoform

*Figure 8 continued on next page*

*Figure 8 continued*

was not efficiently extracted and was hence undetectable. (**E**) As *A*, expect that HEK293T cells were transfected to express HA-tagged ZDHHC9wt or XLID mutant forms from *Figure 2*, plus the indicated additional proteins. Western blots of palmitoyl fractions (isolated by ABE assay) and total proteins are shown. (**F**) Quantified data from *E* for palmitoylation of MBP confirm that palmitoylation of MBP by ZDHHC9-P150S and ZDHHC9wt do not differ significantly and that palmitoylation of MBP by ZDHHC9-R96W is less impaired than that by ZDHHC9-R148W (****$p<0.0001$; ***$p<0.001$; *$p<0.05$; n.s.; not significant, ANOVA with Dunnett's multiple comparison test). (**G**) Quantified data from *E* for autopalmitoylation of ZDHHC9 confirm that ZDHHC9wt and ZDHHC9-P150S autopalmitoylate to broadly similar extents but that autopalmitoylation of both ZDHHC9-R96W and ZDHHC9-R148W is significantly lower than that of ZDHHC9wt (****$p<0.0001$; ***$p<0.001$; *$p<0.05$; n.s.; not significant, ANOVA with Dunnett's multiple comparison test). Data in *B*, *D*, *F*, *G* are mean $\pm$ SEM.

The online version of this article includes the following source data and figure supplement(s) for figure 8:

**Source data 1.** Original western blots for images shown in *Figure 8A, C and E*.

**Source data 2.** PDF showing original western blots for images shown in *Figure 8A*, with relevant conditions and cropped regions marked.

**Source data 3.** PDF showing original western blots for images shown in *Figure 8C*, with relevant genotypes and cropped regions marked.

**Source data 4.** PDF showing original western blots for images shown in *Figure 8E*, with relevant conditions and cropped regions marked.

**Figure supplement 1.** Zdhhc9 loss impacts the palmitoyl-protein myelin-associated glycoprotein (MAG) but not Cadm4.

**Figure supplement 1—source data 1.** Original western blots for images shown in *Figure 8—figure supplement 1*.

**Figure supplement 1—source data 2.** PDF showing original western blots for images shown in *Figure 8—figure supplement 1*, with relevant genotypes and cropped regions marked.

## Discussion

Genetic factors are central to intellectual disability (*Vorstman and Ophoff, 2013*; *Vissers et al., 2016*), as exemplified by the increasing number of known XLID-associated genes (*Renpenning et al., 1962*; *Amir et al., 1999*; *Billuart et al., 1998*; *Neri et al., 2018*). However, knowledge of the cellular/molecular processes controlled by XLID-associated gene products is often limited. The cognitive deficits and epilepsy that are hallmarks of *ZDHHC9*-associated XLID are often ascribed to malformations of neocortical gray matter (*Stevenson et al., 2013*; *Guerrini and Dobyns, 2014*; *Lim and Crino, 2013*; *Zhang et al., 2010*; *Juric-Sekhar and Hevner, 2019*). Indeed, ZDHHC9 is expressed in a subset of forebrain neurons (*Doyle et al., 2008*), and 'Zdhhc9 knockdown' primary hippocampal neurons display impaired dendritic branching and an altered ratio of excitatory-to-inhibitory synapses (*Shimell et al., 2019*). However, there is an increasing appreciation that ID can also result from impaired WM formation and/or function (*Shangguan et al., 2019*; *Wong, 2019*; *Yu et al., 2008*; *Spencer et al., 2005*; *Soldán and Pirko, 2012*). Consistent with this notion, both ZDHHC9 and its partner protein Golga7 are far more highly expressed in OLs than in other CNS cell types, in both mice and humans (*Figure 1E and F*, *Zhang et al., 2014*; *Zeisel et al., 2018*; *Zhang et al., 2016*). In support of a key role for ZDHHC9 in normal WM formation and function, we found that *Zdhhc9* KO does not affect axon number in the CC, but greatly impacts microscale OL morphology and myelin ultrastructure (*Figures 4–6*). We recognize that using conventional *Zdhhc9* KO mice, in which *Zdhhc9* is globally deleted, cannot directly address whether impaired myelination in these mice is due to a cell-autonomous role for ZDHHC9 in OLs, a role for ZDHHC9 in neurons, or a combination of these factors. However, the impaired OL morphology and maturation seen in culture conditions after acute *Zdhhc9* loss (*Figure 7*) supports the first of these possibilities. While a conditional KO mouse could more directly help test the OL-autonomous role of ZDHHC9 in vivo, in this study, we focused on conventional KO mice to more accurately mirror the situation in human patients with *ZDHHC9* loss or mutation.

It is perhaps surprising that, despite the clear changes in OL morphogenesis, we did not detect changes in overall numbers of OPCs or OLs in *Zdhhc9* KO mice. However, we cannot exclude the possibility that such changes could be revealed by different methods. In addition, different OL subtypes express different subsets of other marker genes (*Soldán and Pirko, 2012*) and it remains possible that *Zdhhc9* loss preferentially directs OL maturation towards or away from one or more of these subtypes. Furthermore, we cannot exclude the possibility that assessment of other brain regions might also reveal differences in OPC and/or OL numbers in *Zdhhc9* KO mice. Lastly, we also note that acute *Zdhhc9* loss in cultured OLs causes more striking phenotypes than seen with KO mice in vivo. However, it is not uncommon for acute knockdown to cause more dramatic effects than germline KO, possibly due to longer-term compensatory mechanisms in vivo (*George et al., 2015*; *Elias et al.,*

*2006*). Together, though, our findings suggest that OL and WM abnormalities could significantly contribute to *ZDHHC9*-associated XLID.

We also recognize that although *Zdhhc9* loss affects OL morphology both in vitro and in vivo, the specific nature of these changes differs between the two systems (compare *Figures 4 and 7*). However, several non-mutually exclusive factors could account for these differences. First, morphology in vivo may well be influenced by the axons and/or other extrinsic components around each OL that are not present in our primary cultures. Second, OL growth in vivo is highly 3-dimensional, whereas growth in culture is largely 2-dimensional – it may be difficult to support formation of spheroids (by definition, a 3-dimensional structure) in the latter situation. Finally, ZDHHC9 is absent in vivo from the beginning of development until the time points examined, whereas in our cultured OL experiments, Zdhhc9 shRNA is virally delivered to OPC cultures at DIV2 and likely acutely affects Zdhhc9 expression predominantly in committed OLs (following the switch to differentiation medium at DIV3). These differences may also affect the ability of other PATs or, potentially, palmitoylation-independent subcellular processes, to compensate for Zdhhc9 loss. Further experiments, perhaps involving longitudinal imaging of the development of fluorescently labeled OLs in the two systems, could provide more insights into how *Zdhhc9* loss affects OL morphological elaboration.

Based on behavioral milestones, patients with *ZDHHC9*-associated XLID have been diagnosed as young as two years of age (*Schirwani et al., 2018*), suggesting that ZDHHC9 is important for higher brain function during early development. Consistent with this notion, we observed dysmyelination (both hyper and hypomyelination) in P30 mice, approximately equivalent to a 3-year-old human (*Dutta and Sengupta, 2016*). These findings suggest that the myelin abnormalities observed in adult *Zdhhc9* KO mice (*Figure 6*) are due to impaired initial myelination, rather than impaired myelin maintenance, and further support the use of *Zdhhc9* KO mice as a model for human *ZDHHC9*-associated XLID.

Although difficult to prove causation in a global, conventional KO line, the myelin abnormalities in *Zdhhc9* KO mice are also associated with behavioral impairments that have some overlap with human patients with *ZDHHC9*-associated XLID. Most notably, *Zdhhc9* KO mice have seizures, reminiscent of the rolandic epilepsy reported in many human patients with *ZDHHC9* mutation (*Baker et al., 2015*; *Shimell et al., 2019*). *Zdhhc9* KO mice also have impaired latency to reach a hidden platform in the Morris Water Maze (MWM), a task designed to assess spatial learning (*Kouskou et al., 2018*). However, the KO mice do not differ significantly in reference memory in a subsequent probe trial, suggesting that, despite the increased latency, they do learn the location of the platform as effectively as wt (*Kouskou et al., 2018*). In addition, performance in both the open field and in an elevated plus maze (EPM, in which the KO mice spend significantly more time in the open arms) suggests that *Zdhhc9* KO mice also have decreased levels of anxiety (*Kouskou et al., 2018*). Decreased anxiety is reported in several other mouse models of ID (*Zang et al., 2009*; *Samaco et al., 2008*; *Escorihuela et al., 1998*; *Peier et al., 2000*; *Altafaj et al., 2001*). Finally, *Zdhhc9* KO mice spend significantly less time on a hanging wire task (*Kouskou et al., 2018*), a phenotype consistent with hypotonia, a phenotype also reported in patients with *ZDHHC9* mutation (*Raymond et al., 2007*; *Masurel-Paulet et al., 2014*; *Han et al., 2017*). Like reduced anxiety, hypotonia is seen in several human neurodevelopmental conditions and relevant mouse models (see *Kouskou et al., 2018* and references therein for a fuller discussion). While either OL-specific conditional KO of *Zdhhc9* or OL-specific rescue in a KO background would be required to determine causation. However, these findings are consistent with a model in which the impaired myelination in the absence of *Zdhhc9* causes behavioral abnormalities reminiscent not just of *ZDHHC9* mutation in humans, but also in other forms of ID.

*Zdhhc9* is also expressed at higher levels than other PATs in OLs (*Figure 1C, E and F*), potentially explaining much of the impact of *Zdhhc9* loss in these cells. However, it is also intriguing that ZDHHC9wt localization in OLs differs markedly from that of other PATs examined, and of three ZDHHC9 XLID mutants (*Figure 2*). These findings suggest that specific subcellular localization of ZDHHC9wt is central to the role of this PAT in OLs and explain why other PATs, and ZDHHC9 XLID mutant forms cannot compensate for loss of ZDHHC9wt. This notion is further supported by our findings that the XLID-associated P150S and, to a lesser extent, R96W mutant forms of ZDHHC9 are still capable of palmitoylating MBP in cotransfected cells. Indeed, despite its reduced catalytic activity in an in vitro assay (*Mitchell et al., 2014*), ZDHHC9-P150S does not differ significantly from ZDHHC9wt in its ability to palmitoylate MBP, or to autopalmitoylate, in our cell-based assay. This residual autopalmitoylation, which was also observed in *Mitchell et al., 2014*, suggests that ZDHHC9-P150S might

still be capable of palmitoylating MBP and/or other substrates in OLs, if it were to be targeted to the correct subcellular location. Impaired subcellular targeting thus represents an additional possible contributory factor, not mutually exclusive with impaired catalytic activity, to explain impairments in ZDHHC9-associated XLID. More broadly, these results emphasize the importance of precise subcellular localization in palmitoylation-dependent regulation, a key take home message from prior studies by ourselves and others e.g., (*George et al., 2015*; *Zhang and Thomas, 2024*; *Furne et al., 2006*).

What, then, is the identity of the puncta in OL subcellular processes to which ZDHHC9wt can localize, but which other PATs and ZDHHC9 XLID mutants cannot? Our immunostaining results with subcellular organelle markers suggest that these puncta represent Golgi outposts or satellites, and that ZDHHC9's key partner Golga7 also localizes to this location (*Figure 2*, *Figure 2—figure supplement 2A*; *Govind et al., 2021*; *Fu et al., 2019*). Golgi outposts have long been known in neurons (*Horton and Ehlers, 2003*), and we previously reported that ZDHHC9 localizes to such structures in primary hippocampal neurons (*Shimell et al., 2019*). Golgi outposts were also recently described in OLs (*Wang and Gleeson, 2020*). However, the function of these structures in OLs remains to be fully determined, so the role(s) of ZDHHC9 at this subcellular organelle is challenging to infer. Nonetheless, we speculate that ZDHHC9 could be involved in either or both of the best-described roles of Golgi outposts, protein glycosylation and microtubule nucleation, for the following reasons. First, ZDHHC9-positive Golgi outposts in OL processes are also positive for the enzyme Mannosidase II, suggesting they represent sites at which protein glycosylation is regulated (*Figure 2—figure supplement 2B*). In addition, although ZDHHC9 colocalization with TPPP, a protein implicated in microtubule nucleation, is less apparent (*Figure 2—figure supplement 2C*), knockout/knockdown of either ZDHHC9 (*Figures 4 and 7*) or TPPP (*Fu et al., 2019*) affects branching and complexity of OL processes, a process that involves microtubule nucleation. However, more investigation is needed to determine the functions of Golgi outposts in OLs, in order to define the associated contribution of ZDHHC9.

Another intriguing question is whether and how the loss of ZDHHC9 action at Golgi outposts might contribute to the myelination deficits we observed in *Zdhhc9* KO mice (*Figures 5 and 6*). Our sparse genetic labeling studies revealed that *Zdhhc9* KO OL processes are distended, with numerous spheroid-like swellings (*Figure 5*). We reason that these spheroids are enclosed by the OL lipid membrane because if the membrane were ruptured, the EGFP signal would likely diffuse. This, in turn suggests that the caliber of the OL process at the position of the spheroid is grossly abnormal, i.e., the membrane has hyper-expanded. Given that OL membrane growth during myelination extends in two directions, i.e., spiral growth to the axonal surface and longitudinal growth along the axon, it is possible that spheroid-like structures are formed by uneven myelin growth. We recognize that we cannot yet conclude whether and how spheroid formation might relate to the myelination deficit in Zdhhc9 KO mice that we observe by EM (*Figure 6*). However, it is not inconceivable that the hypo- and hypermyelination of *Zdhhc9* KO axons seen in single-plane EM cross-sections (*Figure 6*) represents this same distension phenotype, observed using a different method. Although beyond our current technical abilities, we speculate that EM-based reconstruction in three dimensions would reveal regions of both hypo- and hypermyelination of individual callosal axons in *Zdhhc9* KO mice.

How, though, might *Zdhhc9* loss cause dysmyelination? Several myelin proteins are palmitoylated, and we found that a ZDHHC9/Golga7 complex palmitoylates the major myelin protein MBP and that MBP palmitoylation is reduced in forebrain WM of *Zdhhc9* KO mice (*Figure 8*). Given that palmitoylation can sort proteins to the myelin membrane (*Schneider et al., 2005*), the simplest explanation might be that impaired palmitoylation of MBP, and potentially other myelin proteins, in the absence of ZDHHC9 affects myelin structure per se. It would be of considerable interest to investigate whether and how *Zdhhc9* loss alters the distribution of specific MBP splice variants. However, the anti-MBP antibody used in this study recognizes all forms of MBP, not just the specific splice variants whose palmitoylation is affected by *Zdhhc9* loss. Specifically assessing nanoscale distribution of these splice variants would require a way (e.g. anti-MBP splice form-specific antibodies that are compatible with immuno-EM) to distinguish these variants from other, non-palmitoylated forms of MBP. Developing such an antibody could greatly facilitate future studies of nanoscale MBP distribution in the presence and absence of ZDHHC9.

We also observed slight but significant reductions in palmitoyl- and total levels of MAG in the absence of ZDHHC9 (*Figure 8—figure supplement 1A and B*), which may also contribute to impaired myelination. In contrast, ZDHHC9 loss did not alter palmitoyl- or total levels of Cadm4, a myelin

protein whose palmitoylation is ascribed to the Golgi-localized PAT ZDHHC3 (*Chang et al., 2022*; *Figure 8—figure supplement 1C and D*). This latter finding suggests that, although ZDHHC9 loss may affect additional myelin proteins, gross, widespread dysregulation of myelin protein levels and/ or palmitoylation in the absence of ZDHHC9 is unlikely. Importantly, several mutant forms of ZDHHC9 fail to localize to Golgi outposts in OLs and lead to XLID, even when showing apparently normal catalytic activity. This latter observation is consistent with a model in which ZDHHC9 action at OL Golgi outposts is critical for normal myelination. An additional factor to consider, however, is that the key ZDHHC9 substrate(s) may not be the myelin proteins themselves, but rather other proteins that act at Golgi outposts to direct myelin proteins to the correct region of the myelin membrane and/or to ensure uniform distribution of myelin proteins within that membrane. Another non-mutually exclusive possibility is that ZDHHC9 acts at Golgi outposts but indirectly, for example, to drive the expression of myelin protein genes. A comparison of WT and *Zdhhc9* KO palmitoyl-proteomes could help determine which of these possibilities (which are not mutually exclusive) best accounts for impaired myelination in the absence of ZDHHC9.

Interestingly, despite their marked differences in basal catalytic activity (as assessed by autopalmitoylation), wt and all XLID forms of ZDHHC9 appear to show enhanced activity (measured by both auto- and MBP palmitoylation) in the presence of Golga7, suggesting that the association with Golga7 (which also localizes to Golgi outposts) is central to ZDHHC9 activity. This model is also highly consistent with the biased expression of Golga7 in OLs, compared to other CNS cell types (*Figure 1E and F*). Moreover, XLID-associated mutant forms of ZDHHC9 also show reduced protein stability and are impaired in their ability to form complexes with Golga7 (also known as Golgi Complex Protein 16 kDa; GCP16)(*Nguyen et al., 2023*). ZDHHC9 XLID mutations may thus affect ZDHHC9 trafficking to Golgi outposts per se, but alternatively, or in addition, may decrease formation/stabilization of ZDHHC9/Golga7 complexes at these locations. Either or both of these mechanisms, which are not mutually exclusive, may contribute to impaired MBP palmitoylation and/or myelination deficits caused by *ZDHHC9* mutation.

In summary, we reveal an unexpectedly non-heterogeneous expression of *Zdhhc9* in both mice and humans, with biased expression in mature OLs. Moreover, within OLs, WT ZDHHC9 localizes uniquely compared to other PATs examined, and to XLID-associated forms of ZDHHC9. Although we did not detect changes in overall OL numbers or gross myelin staining, we found that *Zdhhc9* loss greatly impairs OL morphology and myelination at the microscale. These findings provide new insights into mechanisms of *ZDHHC9*-associated XLID and into other conditions marked by WMI.

# Materials and methods
## Mice and rats

*Zdhhc9* KO mice (B6;129S5-Zdhhc9tm1Lex/Mmucdm) were originally obtained from Mutant Mouse Resource and Research Center (MMRRC), UC Davis, and were previously described (*Shimell et al., 2019*). Mice were transferred from the University of British Columbia to Temple University School of Medicine for this study. Female heterozygous *Zdhhc9* KO mice were bred with male wild-type C57BL/6 mice to obtain male *Zdhhc9* hemizygous KO and male wild-type mice as littermate controls, which were used for experiments in *Figures 6 and 8*. Experiments in *Figures 3–5* used a mixture of male *Zdhhc9* hemizygous KO (*Zdhhc9* $^{y/-}$) and female *Zdhhc9* homozygous KO (*Zdhhc9* $^{-/-}$) mice. *Mobp-EGFP* BAC Tg (*Kang et al., 2013*) (generated by GENSAT; MMRRC stock #030483-UCD, RRID:MMRRC_030483-UCD), *R26-lsl-EGFP* (*RCE*) (*Sousa et al., 2009*) (MMRRC stock #032037-JAX, RRID:MMRRC_032037-JAX) and *Mobp-iCreER* mice (*Heo et al., 2022*) were described previously and obtained from Dr. Dwight Bergles (Johns Hopkins University). *Pdgfra-CreER* BAC Tg (*Kang et al., 2010*; RRID:IMSR_JAX:018280) and *mT/mG* mice (*Muzumdar et al., 2007*) were purchased from the Jackson Laboratory. Timed pregnant Sprague Dawley rats were obtained from Charles River and gave birth in the University Laboratory Animal Resources (ULAR) facility at Temple University School of Medicine. Neonatal rat pups were used for primary cultures as described under 'Rat Primary OPC Culture and Differentiation.' All mice and rats were housed in a barrier facility with a 12 hr:12 hr light: dark cycle, were provided food and water ad libitum and were checked daily by ULAR staff. This study was performed in strict accordance with the recommendations in the Guide for the Care and Use of Laboratory Animals of the National Institutes of Health. All animals were handled according

to approved institutional animal care and use committee (IACUC) protocols (#4939, #4943, #4866) of Temple University.

## Fluorescence-activated cell sorting (FACS)

After the whole brain was isolated from one-month-old *Mobp-EGFP* mice, one hemisphere of the forebrain was chopped with a blade into small pieces. The brain cells were further dissociated using a Neural Dissociation Kit (Miltenyi Biotec) according to the manufacturer's instructions. After enzymatic digestion, the cells were mechanically dissociated with gentle pipetting and suspended in Hank's Balanced Salt Solution (GIBCO). The cell suspension was passed through a 40 μm cell strainer (Corning). Cells were re-suspended in 0.5% FBS in HBSS and isolated with BD Influx (BD Biosciences) at the Flow Core Facility of Temple University School of Medicine. The other hemisphere of each mouse brain was used for total RNA isolation.

## RNA sequencing

Total RNAs were extracted from the FACS-isolated EGFP$^+$ OLs with RNeasy Micro kit (Qiagen). For total RNA extraction from the other hemisphere forebrain (without OL sorting), we homogenized the brains with Trizol reagent (Invitrogen). All RNA samples were subjected to Quality Control with Bioanalyzer (Agilent), and the RNA samples whose RIN was higher than 8 were used for subsequent applications. Ten ng of total RNA was used in the Ovation RNA-Seq System v2 (NuGEN) to prepare cDNA library. Following the manufacturer's instructions, total RNA and primer were incubated at 65 °C for 5 min, followed by first-strand cDNA synthesis (4 °C for 1 min, 25 °C for 10 min, 42 °C for 10 min, 70 °C for 15 min, and then cooling to 4 °C) and second strand cDNA synthesis (4 °C for 1 min, 25 °C for 10 min, 50 °C for 30 min, 80 °C for 20 min and then cooling to 4 °C). RNAClean XP bead purification was performed. Single Primer Isothermal Amplification (SPIA) was used to amplify cDNAs. QIAquick cleanups (Qiagen) were eluted in 30 μl low EDTA TE buffer and quantified via Nanodrop 1000. 100 ng amplified SPIA cDNA was sonicated using a Covaris E210 (50 μl, duty cycle 10%, intensity 5, 200 burst/s, 45 s) to shear samples to 350 bp. The libraries were prepared using the TruSeq DNA LT Sample Prep Kit (Illumina) according to the manufacturer's instructions. Samples were purified using sample purification beads. End repair reaction (at 30 °C for 30 min) was followed by bead purification and size selection for 350 bp insert. dA-tailing (37 °C for 30 min, 70 °C for 5 min, 4 °C for 5 min) and adapter ligation with barcoded adapters were performed (30 °C for 10 min), followed by bead purification and PCR amplification (95 °C for 3 min; 8 cycles of 98 °C for 20 s, 60 °C for 15 s and 72 °C for 30 s; 72 °C for 5 min). A final bead purification was performed, and libraries were quantified using the Agilent Bioanalyzer High Sensitivity DNA assay. For the forebrain total RNA library (all cell RNA-seq), 500 ng of RNA was used for library generation with TruSeq Stranded mRNA Library Prep Kit (Illumina). Sequencing was performed using an Illumina HiSeq 2000.

## RNA-seq data analyses

RNA-seq reads were aligned to a mouse reference genome (mm10) by the STAR alignment tool (v2.4.0) with default options and PCR duplicate reads were removed using the Picard Tools (v1.124). To quantify expression levels for each gene, we counted the number of aligned fragments for each gene using HTSeq-0.6.1 with parameters (-s no, -r pos, -f bam, -m intersection-nonempty and -t exon) according to the Ensembl mouse transcript annotation (GRCm38.74 version) and calculated the FPKM (Fragments Per Kilobase per Million mapped reads) values of each gene. For heatmap visualization, the FPKM values for each gene were color-coded with the Microsoft Excel color scales.

## Accession codes

The RNA-seq files have been uploaded to the European Nucleotide Archive (ENA) under accession code PRJEB19341.

## Tamoxifen administration

Cre activity was induced with tamoxifen (Sigma-Aldrich, Cat# T5648) administration to *Pdgfra-CreER* mice. Tamoxifen was dissolved (20 mg/ml) in a mixture of sunflower seed oil-ethanol (10:1), and then ethanol was evaporated in a vacuum concentrator for 30 min. Forty mg/kg (b.w.) of tamoxifen was intraperitoneally (i.p.) injected twice daily with at least a 6 hr interval between injections. A total of

eight doses of tamoxifen was injected into the *Pdgfra-CreER mice; RCE; ±Zdhhc9* KO mice between P21 and P24.

## Rat primary OPC culture and differentiation

Primary mixed glial cultures were prepared from P1 rat pups, as previously described (*Sánchez-Gómez et al., 2018*). Briefly, cortices were isolated and digested with papain and DNase I, followed by mechanical dissociation. Cells were resuspended in DMEM supplemented with 10% (v/v) fetal bovine serum and 1% (w/v) penicillin/streptomycin. Cells were plated in T75 flasks coated with poly-D-lysine and medium was replaced every other day. Under these conditions, a mixed population of glial cells survives and proliferates, but neuronal cells do not survive. After 7-10 days, these mixed glial cultures were shaken overnight (14-16 hr). Detached cells were added to uncoated petri dishes, to which microglia preferentially adhere but OPCs do not. The OPC-enriched supernatant was then plated on poly-D-lysine-coated coverslips. OPCs were maintained in defined OPC media containing PDGF (10 ng/ml) and bFGF (5 ng/ml). The following day, fresh OPC medium was added, and the medium was refreshed every other day. To induce OPC differentiation, OPC medium was replaced by defined OL medium containing triiodothyronine (T3, 30 ng/ml and T4, 40 ng/ml), CNTF (10 ng/ml), and NT3 (1 ng/ml).

## Molecular biology

Wild-type mouse *Zdhhc3*, *Zdhhc7*, *Zdhhc9*, and *Zdhhc17* cDNAs (all with N-terminal HA tag) were a kind gift from Dr. Masaki Fukata (National Institute of Physiological Sciences, Okazaki, Japan) and were subcloned into lentiviral expression vector (FEW) downstream of the EF1α promoter and an N-terminal HA tag, as described (*Collura et al., 2020*). XLID mutant forms of *Zdhhc9* (human point mutations introduced into mouse cDNA) were generated as *BsrGI – AfeI* gBlock fragments (Integrated DNA Technologies) and were used to replace the wild-type *Zdhhc9 BsrGI – AfeI* fragment *via* standard subcloning. Mouse *Golga7* and MBP (21.5 kDa isoform) cDNAs were synthesized as *XhoI-NotI* gBlock fragments and subcloned into a modified FEW vector downstream of an N-terminal myc epitope tag. Mouse *Tppp* cDNA was synthesized (Genewiz) and subcloned into modified FEW vector upstream of a C-terminal Flag tag. C-terminally GFP-tagged mannosidase cDNA was obtained from Addgene (plasmid #160905) and used without additional subcloning. Membrane-bound GFP (mGFP) cDNA was generated by appending the N-terminal 40 amino acids of MARCKs to eGFP (*Laux et al., 2000*) as a *PspXI – HindIII* fragment, which was then subcloned into FEW vector with no additional tag.

## Lentiviral vectors and lentivirus preparation

VSV-G pseudotyped lentiviruses were prepared as described (*Thomas et al., 2012*), except that viruses were collected in DMEM containing 0.1% (v/v) FBS and were added to cultured cells without ultracentrifugation.

## Lentiviral infection and OL differentiation

For *Zdhhc9* knockdown, OPCs were infected at two days in vitro (DIV2) with lentiviral vectors carrying *Zdhhc9* shRNA or a control scrambled shRNA (*Shimell et al., 2019*). OPC media was refreshed at 4 hr and 1 day post-infection, and then the medium was replaced with defined OL medium at 1 day post-infection for differentiation. The OL culture medium was then refreshed every other day. Infected OLs were fixed at 9 DIV ('OL 6 days') for immunostaining.

## Transfection of cultured OPCs

To define the subcellular localization of ZDHHC PATs and other proteins in OLs, differentiated OLs at 6 DIV ('OL 3 days') were transfected with plasmids using Lipofectamine LTX with Plus Reagent (Thermo Fisher Scientific) according to the manufacturer's recommendations. Briefly, plasmid DNA (diluted in Opti-MEM, GIBCO, Thermo Fisher Scientific, Waltham, MA) was mixed with 1.5 µl of PLUS Reagent and incubated for 5 min at room temperature (RT). 1 µl of Lipofectamine LTX (diluted in opti-MEM) was added to pre-incubated DNA and incubated for 30 min at RT for generation of DNA-lipid complex. This DNA-lipid complex was applied to the cells for transfection, and OLs were fixed 9 hr later and processed for immunostaining.

## Immunocytochemistry (ICC)

Coverslips containing OL cells were rinsed with 1 x Recording buffer (25 mM HEPES pH 7.4, 120 mM NaCl, 5 mM KCl, 2 mM $CaCl_2$, 1 mM $MgCl_2$, 30 mM Glucose) and cells were fixed for 20 min in PBS containing 4% paraformaldehyde (PFA)/4% sucrose. Coverslips were washed 3X 10 min in phosphate-buffered saline (PBS) and blocked for 1 hr at room temperature with 10% (v/v) normal goat serum (NGS; Thermo Fisher Scientific, Waltham, MA) in PBST containing 0.15% Triton X-100. Cells were then incubated in blocking solution with primary antibodies overnight at 4 °C, washed 3X 10 min in PBS, incubated in secondary antibodies in blocking solution for 1 hr at RT, washed 3X 10 min in PBS, and mounted on microscope slides using FluorSave reagent (Millipore Sigma). Confocal images were captured with a laser scanning confocal microscope (Leica TCS SP8) and LAS X software.

## Quantification of ICC images

To quantify distribution of HA-tagged PATs in OL processes (*Figure 2*), confocal images of HA (PAT), and GFP (cell fill) signal were thresholded. The same threshold values for each channel were used across all images. Using the Sync Windows tool in ImageJ/Fiji, the cell body area was traced in the GFP channel image and deleted from both images. Both images were then inverted and the AND function was used to generate an image of the $HA^+/GFP^+$ signal. Using the Analyze particles tool (with settings 0-infinity pixels sq and 0.00–1.00 circularity), the combined area occupied by the $HA^+/GFP^+$ puncta was calculated and normalized to the total extra-somatic $GFP^+$ area from the $GFP^+$ image of the same cell.

The percentage of $GFP^+/MBP^+$ cells (*Figure 7B*) was counted manually without thresholding, using the same criteria for all conditions. Sholl analysis in *Figure 7E and F* was performed by manually reconstructing the morphology of individual OLs in ImageJ/Fiji as a binarized image. OL nuclei were considered the center, and concentric circles were drawn with an interval of 10 µm and the number of intersections calculated using the ImageJ/Fiji Sholl Analysis plug-in.

The percentage of morphologically immature OLs (*Figure 7—figure supplement 1*) was calculated manually based on identification of $GFP^+/CNP^+$ cells (committed OLs) that did not show the 'pancake'-like morphology of mature OLs and plotted as a percentage of all $GFP^+/CNP^+$ cells per field. Morphologically immature OLs also typically lacked MBP staining (as in *Figure 7B*) but this property was not used in the *Figure 7—figure supplement 1* analysis.

## Immunofluorescence of brain sections

Mice were anesthetized with pentobarbital sodium (70 mg/kg, i.p.) and transcardially perfused with PBS followed by 4% PFA. Mouse brains were post-fixed overnight at 4 °C and transferred to 30% sucrose in PBS at 4 °C. Brain tissues were frozen in Tissue-Tek optimum cutting temperature (O.C.T.) compound (Sakura, Cat# 4586) and sectioned using a Leica Biosystems CM1950 Cryostat. Three different thicknesses of brain or spinal cord sections were used: 20 µm for EGFP-based or ASPA-staining dependent OL quantification, 35 µm for OPC fate analysis, and 50 µm for mEGFP-based OL morphological analysis. Brain and spinal cord sections were stained in a free-floating manner. Sections were permeabilized with 0.3% (w/v) Triton X-100 and blocked with blocking solution (5% (v/v) normal donkey serum, 0.3% (w/v) Triton X-100) for 1 hr at RT. Sections were then incubated in blocking solution containing primary antibodies at 4 °C overnight. On the next day, sections were rinsed with PBS 3X 5 min and incubated with secondary antibodies and DAPI (1:1000) in blocking solution at RT for 2 hr. Sections were rinsed in PBS 3X 5 min and then mounted onto slide glasses with a mounting medium with ProLong antifade (Thermo Fisher Scientific, P36970). Image acquisition of the immunostained brain sections was performed with a wide-field fluorescent microscope AxioImager M2 (Zeiss) or a laser scanning confocal microscope TCS SP8 (Leica). Images were obtained and analyzed by separate investigators blinded to genotype.

## Morphological analysis of oligodendrocytes

The settings for OL tracing were kept the same for all the samples across genotypes. Confocal images were captured for sparsely labeled randomly chosen $EGFP^+$ OLs from the CTX (layers IV–VI). Images of labelled cells were imported into Fiji and traced without z-projecting the stacks. Sholl analysis was performed using the Fiji plug-in. For Sholl analysis of OLs in vivo, the OL nuclei were considered the

center, and concentric circles were drawn with an interval of 5 µm. Imaris 9.9 (Oxford Instrument) software was used for 3D surface rendering of OLs with representative confocal images.

## Electron microscopy and *g*-ratio analysis

Mice were anesthetized with pentobarbital sodium (100 mg/kg, i.p) and transcardially perfused with PBS followed by 2.5% PFA, 2% glutaraldehyde (in 0.1 M phosphate buffer, pH 7.4). Mouse brains were post-fixed overnight at 4 °C and transferred to 0.1 M phosphate buffer at 4 °C. Brains were dehydrated using a graded ethanol series and embedded into Embed-812 (EMS). Thin sections were prepared, stained with uranyl acetate and lead citrate, and visualized with an electron microscope (JEOL 1010 electron microscope fitted with a Hamamatsu digital camera) at the University of Pennsylvania Electron Microscopy Resource Laboratory. 15,000 X magnified images and Fiji Plug-in and ImageJ/Fiji (*Schindelin et al., 2012*) were used for the g-ratio analysis of myelin. To analyze potentially hypermyelinated small axons in P56 mice, g-ratio was calculated using the ratio of inner-to-outer diameter (distance values derived from pixel intensity along plot profile generated in ImageJ/Fiji) of each myelinated axon. To analyze myelination of all axons in P30 mice, inner and outer diameters of axons were traced manually in ImageJ/Fiji and the resultant diameters calculated.

## Acyl biotinyl exchange assay

HEK293T cells were transfected using a calcium phosphate-based method as described (*Thomas et al., 2012*). The identity of this cell line was authenticated by ATCC using STR profiling and was found to be an exact match to CRL-3216 (HEK293T) in the ATCC STR database. The cells were tested for mycoplasma and were found to be negative. Cells were lysed 8 hr post-transfection and ABE assays performed as in *Thomas et al., 2012*. For WM tissue collection from *Zdhhc9* knockout and littermate control mice, mice were anesthetized with pentobarbital sodium (100 mg/kg, i.p). Brains were isolated and briefly rinsed with ice-cold HBSS buffer (no calcium, no magnesium). Using a Mouse Brain Slicer, 1 mm block coronal brain sections were prepared on ice and WM-enriched tissues, including corpus callosum and striatum were dissected from slices under a dissection microscope. Dissected WM tissue was homogenized in a glass-teflon dounce homogenizer (20 strokes, 200 rpm) in 4 mM HEPES, 0.32 M sucrose, containing freshly diluted Protease Inhibitor Cocktail (PIC; Roche). Homogenized samples were transferred to a fresh tube, rapidly brought to room temperature, and solubilized by addition of 1/10 volume of 10% (w/v) SDS. Samples were centrifuged for 10 min at 4 °C at 13,000 rpm to pellet insoluble material and protein concentrations in supernatants determined by BCA assay (Pierce). Protein concentrations were normalized by addition of homogenization buffer containing SDS and PIC and were used for ABE assay as described (*Thomas et al., 2012*).

## Quantification and statistical analysis

All data were analyzed using GraphPad Prism software (GraphPad Software, San Diego, CA). In all graphs, the mean is shown, and error bars indicate standard error of the mean (SEM). For the quantitative comparisons of OL lineage cells, OPC fate analysis, and OL process complexity, two-way ANOVA, and Šidák's multiple comparison test were used. For spheroid structure comparison, an unpaired student's t-test was used.

## Antibodies

Antibodies used in this study, together with RRID and dilution information, are described in the key resources table.

## Acknowledgements

We thank P Kanuparthi, L Hernandez, N Hesketh, and S Yungblut (all Thomas lab) for help with molecular biology, ABE assays, EM image acquisition, and g-ratio quantification, respectively. We also gratefully acknowledge Dr. M Fukata for PAT cDNAs and Dr. D Bergles for Mobp-iCreER mice and anti-NG2 antibody. Supported by grants from NINDS (R21 NS125202-01 to GMT; R01 NS089586 to SHK), Ellison Medical Foundation (AG-NS-1101–13 to SHK), and Shriners' Childrens (87400PHI to GMT).

## Additional information

### Funding

| Funder | Grant reference number | Author |
|---|---|---|
| National Institute of Neurological Disorders and Stroke | NS125202-01 | Gareth M Thomas |
| National Institute of Neurological Disorders and Stroke | NS089586 | Shin H Kang |
| Shriners Hospitals for Children | 87400PHI | Gareth M Thomas |
| Ellison Medical Foundation | AG-NS-1101-13 | Shin H Kang |

The funders had no role in study design, data collection and interpretation, or the decision to submit the work for publication.

### Author contributions

Hey-Kyeong Jeong, Data curation, Formal analysis, Investigation, Methodology, Writing – original draft; Estibaliz Gonzalez-Fernandez, Ilan Crawley, Julia M Coakley, Data curation, Formal analysis, Investigation; Jinha Hwang, Data curation, Formal analysis; Dale DO Martin, Investigation, Methodology; Shernaz X Bamji, Resources, Funding acquisition; Jong-Il Kim, Supervision; Shin H Kang, Conceptualization, Resources, Data curation, Supervision, Investigation, Writing – original draft, Funding acquisition, Writing – review and editing; Gareth M Thomas, Conceptualization, Resources, Data curation, Formal analysis, Supervision, Funding acquisition, Investigation, Writing – original draft, Project administration, Writing – review and editing

### Author ORCIDs

Shernaz X Bamji (ID) https://orcid.org/0000-0003-0102-9297
Jong-Il Kim (ID) https://orcid.org/0000-0002-7240-3744
Shin H Kang (ID) https://orcid.org/0000-0002-3692-9802
Gareth M Thomas (ID) https://orcid.org/0000-0003-3183-8431

### Ethics

This study was performed in strict accordance with the recommendations in the Guide for the Care and Use of Laboratory Animals of the National Institutes of Health. All of the animals were handled according to approved institutional animal care and use committee (IACUC) protocols (#4866, #4939, #4943) of Temple University.

Reviewer #1 (Public review): https://doi.org/10.7554/eLife.97151.3.sa1
Author response https://doi.org/10.7554/eLife.97151.3.sa2

## Additional files

### Supplementary files

MDAR checklist

### Data availability

RNA-Seq data files have been uploaded to the European Nucleotide Archive (ENA) under accession code PRJEB19341. All other data generated or analysed during this study are included in the manuscript and supporting files.

The following dataset was generated:

| Author(s) | Year | Dataset title | Dataset URL | Database and Identifier |
|---|---|---|---|---|
| Hwang J, Kim J, Kang SH | 2019 | Age-dependent changes in translational expression of oligodendrocytes | https://www.ebi.ac.uk/ena/browser/view/PRJEB19341 | European Nucleotide Archive, PRJEB19341 |

The following previously published datasets were used:

| Author(s) | Year | Dataset title | Dataset URL | Database and Identifier |
|---|---|---|---|---|
| Zhang Y, Chen K, Sloan SA, Bennett ML, Scholze AR, O'Keeffe S, Phatnani HP, Guarnieri P, Caneda C, Ruderisch N, Deng S, Liddelow SA, Zhang C, Daneman R, Maniatis T, Barres BA, Wu JQ | 2014 | Mouse data | https://brainrnaseq.org/wp-content/uploads/2022/09/fe-wp-dataset-120.csv | Brain RNA-Seq, Musmusculus.csv |
| Zhang Y, Chen K, Sloan SA, Bennett ML, Scholze AR, O'Keeffe S, Phatnani HP, Guarnieri P, Caneda C, Ruderisch N, Deng S, Liddelow SA, Zhang C, Daneman R, Maniatis T, Barres BA, Wu JQ | 2014 | Human data | https://brainrnaseq.org/wp-content/uploads/2022/09/fe-wp-dataset-124.csv | Brain RNA-Seq, Homosapiens.csv |

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

# Appendix 1

**Appendix 1—key resources table**

| Reagent type (species) or resource | Designation | Source or reference | Identifiers | Additional information |
|---|---|---|---|---|
| Genetic reagent (*Mus musculus*) | B6;129S5-Zdhhc9tm1Lex/Mmucd | PMID:31747610 | MMRRC:032714-UCD | Extensively back-crossed against C57Bl/6 |
| Genetic reagent (*Mus musculus*) | *Mobp-EGFP* BAC Tg | MMRRC and Dr. D Bergles | GENSAT; RRID:MMRRC_030483-UCD | |
| Genetic reagent (*Mus musculus*) | *Mobp-iCreER* | Dr. D Bergles | | |
| Genetic reagent (*Mus musculus*) | *Pdgfra-CreER* BAC Tg | Jackson Laboratory | Strain #:018280, RRID:IMSR_JAX:018280 | |
| Genetic reagent (*Mus musculus*) | *ROSA26-CAG-LSL-EGFP (RCE)* | MMRRC and Dr. D. Bergles | MMRRC:032037-JAX, RRID:MMRRC_032037-JAX | |
| Genetic reagent (*Mus musculus*) | *ROSA26-LSL-EGFP* (mT/mG) | Jackson Laboratory; PMID:11299042 | MMRRC:032037-JAX, RRID:MMRRC_032037-JAX | |
| Antibody | Anti-HA-Tag (Rb monoclonal IgG) | Cell Signaling Technology | #3724 (C29F4) RRID:AB_1549585 | 1:100 (ICC) |
| Antibody | anti-HA.11 Epitope Tag (mouse IgG1) | Covance/Biolegend | HA.11 #MMS-101P RRID:AB_2314672 | 1:100 (ICC), 1:5000 (western blot) |
| Antibody | Anti-GFP (rabbit IgG) | ProteinTech | # 50430–2-AP, RRID:AB_11042881 | 1:500 (IF) |
| Antibody | Anti-GFP (rabbit IgG) | Invitrogen | #A11122 RRID:AB_221569 | 1:1000 (ICC) |
| Antibody | Anti-myc (mouse IgG1) | Cell Signaling Technology | #92013 (E7F9B) RRID:AB_2800176 | 1:200 (ICC) |
| Antibody | Anti-MBP (chicken IgY) | Aves Lab | #MBP RRID:AB_2313550 | 1:1000 (ICC) |
| Antibody | Anti-CNP (rabbit IgG) | Phosphosolutions | #325-CNP RRID:AB_2492062 | 1:1000 (ICC) |
| Antibody | Anti-APC (CC1) Mouse monoclonal IgG2b | EMD Millipore | #OP80 RRID:AB_2057371 | 1:50, with antigen retrieval |
| Antibody | anti-BCAS1 (rabbit IgG) | Synaptic Systems | # 445 003, RRID:AB_2864793 | 1:300 (IF) |
| Antibody | anti-BCAS1 (guinea pig IgG) | Synaptic Systems | # 445 004, RRID:AB_2905591 | 1:300 (IF) |
| Antibody | anti-ASPA (Rabbit IgG) | GeneTex | #GTX113389 RRID:AB_2036283 | 1:500, with antigen retrieval |
| Antibody | Anti-GFP (goat IgG) | Rockland | #600-101-215 RRID:AB_218182 | 1:500 (IF) |
| Antibody | Anti-MBP (mouse monoclonal IgG) | Biolegend | #808401 | 1:700 (IF) |
| Antibody | Anti-MBP (rabbit monoclonal IgG) | Cell Signaling Technology | #78896 (D8X4 Q) RRID:AB_2799920 | 1:700 (IF), 1:500 (western blot) |
| Antibody | anti-NG2 (guinea pig IgG) | Gift from Dr. D. Bergles | PMID:23542689 | 1:4000 (IF) |
| Antibody | Anti-Olig2 (goat IgG) | R&D | #AF2418 RRID:AB_2157554 | |
| Antibody | Anti-MAG (rabbit IgG) | Cell Signaling Technology | #9043 RRID:AB_2665480 | 1:500 (western blot) |
| Antibody | Anti-Myc (rabbit IgG) | Cell Signaling Technology | #2278 (71D10) RRID:AB_490778 | 1:500 (western blot) |

*Appendix 1 Continued on next page*

*Appendix 1 Continued*

| Reagent type (species) or resource | Designation | Source or reference | Identifiers | Additional information |
|---|---|---|---|---|
| Antibody | Anti-Cadm4 (mouse monoclonal IgG) | Antibodies Inc. | #75–247 RRID:AB_10676101 | 1:100 (western blot) |
| Antibody | anti-PDGFRa (goat IgG) | R&D Systems | #AF1062, RRID:AB_2236897 | 1:500 (IF) |
| Antibody | Alexa Fluor 488-conjugated anti-rabbit (donkey IgG) | Jackson ImmunoResearch | # 711-545-152, RRID:AB_2313584 | 1:500 (IF) |
| Antibody | Cy3-conjugated anti-rabbit | Jackson ImmunoResearch | # 711-165-152, RRID:AB_2307443 | 1:500 (IF) |
| Antibody | Alexa Fluor 647-conjugated anti-rabbit (donkey IgG) | Jackson ImmunoResearch | # 711-605-152, RRID:AB_2492288 | 1:500 (IF) |
| Antibody | Cy3-conjugated anti-goat (donkey IgG) | Jackson ImmunoResearch | # 705-165-147, RRID:AB_2307351 | 1:500 (IF) |
| Antibody | Alexa Fluor 647-conjugated anti-goat (donkey IgG) | Jackson ImmunoResearch | # 705-605-147, RRID:AB_2340437 | 1:500 (IF) |
| Antibody | Cy3-conjugated anti-mouse | Jackson ImmunoResearch | # 715-165-151, RRID:AB_2315777 | 1:500 (IF) |
| Antibody | Alexa Fluor 647-conjugated anti-mouse (donkey IgG) | Jackson ImmunoResearch | # 715-605-151,RRID:AB_2340863 | 1:500 (IF) |
| Antibody | Alexa Fluor 647-conjugated anti-guinea pig (donkey IgG) | Jackson ImmunoResearch | # 706-605-148,RRID:AB_2340476 | 1:500 (IF) |
| Antibody | Donkey anti-rabbit IgG, HRP-linked | Jackson ImmunoResearch | #711–0350152 | WB (1:5000) |
| Antibody | AlexaFluor 488 goat anti-chicken polyclonal | Thermo Fisher Scientific | #A-11039 (RRID:AB_142924) | 1:500 (IF) |
| Antibody | AlexaFluor 488 goat anti-rabbit polyclonal | Thermo Fisher Scientific | #A-11032 (RRID:AB_2534091) | 1:500 (IF) |
| Antibody | AlexaFluor 568 goat anti-rabbit polyclonal | Thermo Fisher Scientific | #A-11011 (RRID:AB_143157) | 1:500 (IF) |
| Antibody | AlexaFluor 568 goat anti-IgG2a polyclonal | Thermo Fisher Scientific | #A-21134 (RRID:AB_2535773) | 1:500 (IF) |
| Antibody | AlexaFluor 647 goat anti-IgG2a polyclonal | Thermo Fisher Scientific | #A-21241 (RRID:AB_141698) | 1:500 (IF) |
| Antibody | AlexaFluor 647 goat anti-IgG1 polyclonal | Thermo Fisher Scientific | #A-21240 (RRID:AB_141658) | 1:500 (IF) |
| Antibody | AlexaFluor 647 goat anti-IgG2b polyclonal | Thermo Fisher Scientific | #A-21242 (RRID:AB_2535811) | 1:500 (IF) |
| Chemical compound, drug | MMTS | Thermo Fisher Scientific | #23011 | |
| Chemical compound, drug | Hydroxylamine | Thermo Fisher Scientific | #26103 | |
| Chemical compound, drug | Biotin-HPDP | Soltec Ventures | #B106 | |
| Other | LTX with Plus reagent | Thermo Fisher Scientific | #15338100 | See Materials and methods 'Transfection of Cultured OPCs' |
| Other | High capacity neutravidin-conjugated beads | Thermo Fisher Scientific | #29202 | See Materials and methods 'Acyl Biotinyl Exchange Assay' |
| Other | Published RNA-Seq dataset – mouse CNS cell types | PMID:25186741 | | See also brainrnaseq.org |

*Appendix 1 Continued on next page*

*Appendix 1 Continued*

| Reagent type (species) or resource | Designation | Source or reference | Identifiers | Additional information |
|---|---|---|---|---|
| Other | Published RNA-Seq dataset – human CNS cell types | PMID:26687838 | | See also brainrnaseq.org |
| Recombinant DNA reagent | pMDLg | Addgene | Cat #12251 (RRID:Addgene_12251) | Lentiviral plasmid Gag and Pol expressing plasmid |
| Recombinant DNA reagent | pRSV-Rev | Addgene | Cat #12253 (RRID:Addgene_12253) | Lentiviral Rev expressing plasmid |
| Recombinant DNA reagent | pMD2.G | Addgene | Cat #12259 (RRID:Addgene_12259) | Lentiviral VSV-G envelope expressing plasmid |
| Recombinant DNA reagent | FEGW-Zdhhc9 sh | This study (Thomas Lab) | | Lentiviral plasmid to transduce OPCs and express eGFP plus Zdhhc9 shRNA sequence from PMID:31747610 |
| Recombinant DNA reagent | FEGW | PMID: 26719418 (Thomas Lab) | | Lentiviral plasmid to transduce OPCs and express eGFP |
| Sequence-based reagent | shRNA sequence matching rat and mouse *Zdhhc9* | PMID:31747610 | | Subcloned into FEGW-Zdhhc9 sh vector, described above |
| Sequence-based reagent | MOBP-EGFP-BAC forward primer for genotyping | This study (Kang lab) | MOBP-EGFP (F) | Sequence: TTACTTGCCATAGCCGTTCC |
| Sequence-based reagent | MOBP-EGFP-BAC reverse primer for genotyping | This study (Kang lab) | MOBP-EGFP (R) | Sequence: GAACTTCAGGGTCAGCTTGC |
| Sequence-based reagent | MOBP-iCreER forward primer for genotyping | This study (Kang lab) | MOBP-iCreER (F) | Sequence: GTCCATCCCTGAAATCATGC |
| Sequence-based reagent | MOBP-iCreER reverse primer for genotyping | This study (Kang lab) | MOBP-iCreER (R) | Sequence: AGGATCTCTAGCCAGGCACA |
| Sequence-based reagent | Pdgfra-CreER forward primer for genotyping | This study (Kang lab) | PDGFRa ex2 (F) | Sequence: TCAGCCTTAAGCTGGGACAT |
| Sequence-based reagent | Pdgfra-CreER reverse primer for genotyping | This study (Kang lab) | Cre (R) | Sequence: ATGTTTAGCTGGCCCAAATG |
| Sequence-based reagent | ROSA26-EGFP (RCE) forward primer for genotyping | Dr. D Bergles | RCE-Rosa1 (F) | Sequence: CCCAAAGTCGCTCTGAGTTG TTATC |
| Sequence-based reagent | ROSA26-EGFP (RCE) reverse primer for genotyping | Dr. D Bergles | RCE-Rosa1 (R) | Sequence: GAAGGAGCGGGAGAAATGGA TATG |
| Sequence-based reagent | ROSA26-EGFP (RCE) reverse primer for genotyping | Dr. D Bergles | RCE-CAG (R) | Sequence: CCAGGCGGGCCATTTACCGT AAG |
| Sequence-based reagent | ROSA26-mGFP (mT/mG) forward primer for genotyping | This study (Kang lab) | ROSA-M (F) | Sequence: CTCTGCTGCCTCCTGGCTTCT |
| Sequence-based reagent | ROSA26-mGFP (mT/mG) reverse primer for genotyping | This study (Kang lab) | ROSA-T (R) | Sequence: CGAGGCGGATCACAAGCAATA |
| Sequence-based reagent | ROSA26-mGFP (mT/mG) reverse primer for genotyping | This study (Kang lab) | ROSA-CAG (R) | Sequence: TCAATGGGCGGGGGTCGTT |
| Sequence-based reagent | ZDHHC9 KO forward primer for genotyping | This study (Kang lab) | DNA196-5 (ZDH9-F) | Sequence: GAAAGAAGGTGACACGGAAA TG |
| Sequence-based reagent | ZDHHC9 KO reverse primer for genotyping | This study (Kang lab) | DNA196-6 (ZDH9-R) | Sequence: CAAATGCCCAGGAGGTACTGT |
| Sequence-based reagent | ZDHHC9 KO forward primer for genotyping | This study (Kang lab) | Neo 2 (F) | Sequence: CGATGCCTGCTTGCCGAATA |
| Software, algorithm | Fiji | PMID:22743772 | RRID:SCR_002285 | https://imagej.net/software/fiji/ |
| Software, algorithm | GraphPad Prism | GraphPad Software | RRID:SCR_002798 | https://www.graphpad.com/ |

