## [Editor Report · eLife Assessment]

This study provides an in-depth exploration of the impact of X-linked ZDHHC9 gene mutations on cognitive deficits and epilepsy, with a particular focus on the expression and function of ZDHHC9 in myelin-forming oligodendrocytes (OLs). These **valuable** findings offer insights into ZDHHC9-related X-linked intellectual disability (XLID) and shed light on the regulatory mechanisms of palmitoylation in myelination. The experimental design and analysis of results are **solid**, providing a reference for further research in this field.

---

## [Referee Report · Reviewer #1 (Public review)]

Summary:

Having shown that acyltransferase ZDHHC9 expression is far higher in myelinating oligodendrocytes (OLs) than in other CNS cell types, Jeong and colleagues focus on exploring the role of ZDHHC9 in myelinating OLs in particular in the palmitoylation of several myelin proteins. This study is relevant in the context of X-linked intellectual disability as it suggests a more relevant role for myelinating glia than previously thought. It also provides useful insights the mechanisms of ZDHHC9-associated XLID and on the palmitoylation-dependent control of myelination.

Strengths:

Well written paper

In general good data quality

Use of transgenics strategies (in addition to the ZDHHC9 KO) strengthen the data and claims

Weaknesses:

A few claims might have needed better experimental support but new data and revised discussion sections addressed some of these weaknesses

---

## [Author Response]

The following is the authors’ response to the original reviews

**Public Reviews:**

**Reviewer #1 (Public Review):**
In this work Jeong and colleagues focus on exploring the role of the acyltransferase ZDHHC9 in myelinating OLs in particular in the palmitoylation of several myelin proteins. After confirming the specific enrichment of the Zdhhc9 transcript in mouse and human OLs, the authors examine the subcellular localization of the protein in vitro and observed that in comparison with other isoforms, ZDHHC9 localizes at OLs cell bodies and at discrete puncta in the processes. These observations (Figures 1 and 2) led the authors to hypothesize that ZDHHC9 plays an important role in myelination. No gross changes were detected in OL development in Zdhhc9 KO mice and analyses from P28 Zdhhc9 KO mice crossed with Mobp-EGFP reporter mice did not show changes in EGFP+ OL differentiation (Figure 3).However, and given the observed subcellular localization of ZDHHC9 in OL processes (Figure 2) and the observation that the percentage of unmyelinated axons is increased in Zdhhc9 KO (Figure 6), early time points to examine the differentiated pools of OLs and their capacity to extend processes/contact axons need to be considered.

We appreciate this point, but due to the order in which experiments were performed, the ZDHHC9 KO mouse colony that we maintained after initial submission of this work contains homozygous MOBP-EGFP, but not the mT/mG transgene that would be most optimal for the proposed experiment. We hope the reviewer appreciates that it would take considerable time and effort regarding mouse breeding to cross out the MOBP and add back the mT/mG. We nonetheless appreciate the importance of the point raised and therefore examined an earlier developmental time point (P21, 3 weeks) to quantify OLs and NG2+ OPCs. In our updated Fig 3C1-C3, we use Mobp-EGFP mice to show that Zdhhc9 KO does not significantly affect the number of EGFP+ OLs at this time point in the cortex, corpus callosum and spinal cord. We also show that in corpus callosum, Zdhhc9 KO does not significantly affect the number of NG2+ OPCs at this earlier time point (Fig 3D, E). Furthermore, immunostaining to detect BCAS1, a marker of pre-mature OLs, also revealed no qualitative difference with ZDHHC9 loss at P21. We show representative images from these BCAS1 experiments in an updated Fig S3. While these new experiments do not address the morphology of OLs in Zdhhc9 KO, they do provide further evidence that deficits in myelination in young Zdhhc9 KO mice (Figure 6) are not likely due to gross differences in OPC or OL numbers during development.

Maturation of OL in Zdhhc9 KO was examined by crossing Zdhhc9 KO with Pdgfra-CreER;R26- EGFP and following the newly EGFP-labelled OPCs following tamoxifen administration. No changes in the numbers of EGFP+ OL were detected. The authors concluded that the loss of ZDHHC9 does not alter oligodendrogenesis in either the young or mature CNS. The authors observed defects in Zdhhc9 KO OL protrusions that they attributed to abnormal OL membrane expansion (Fig 4 and 5). Can they show evidence for this?

This is an important point, and we appreciate the opportunity to explain the reasoning behind our initial statement more fully, while noting that other explanations are possible. Fig 5B (an Imaris-assisted reconstruction using the EGFP cell fill/morphology marker) highlights large spheroid-like distensions along OL processes. We reason that these spheroids are enclosed by the OL lipid membrane because if the membrane were ruptured, the EGFP signal would likely diffuse. This in turn suggests that the caliber of the OL process at the position of the spheroid is grossly abnormal i.e. the membrane has hyper-expanded. Given that OL membrane growth during myelination extends in two directions, i.e., spiral growth to the axonal surface and longitudinal growth along the axon, it is possible that spheroid-like structures are formed by uneven myelin growth. We recognize that we cannot yet conclude whether and how spheroid formation might be linked to the myelination deficit that we observe in Zdhhc9 KO mice. However, defining the subcellular mechanism for spheroid formation may provide further insights into this issue. We have therefore largely retained the original statement but have added the reasoning above to our revised Discussion.

The authors report that Zdhhc9 KO primary and secondary branches in OL were longer, some contained spheroid-like swellings and the OL protrusion complexity was higher. However, these data is partially contradictory to what they show in OL differentiation experiments in vitro (Fig 7). There is also no evidence for increased membrane expansion in Zdhhc9 knockdown myelin forming cells in culture. How to reconcile this?

We appreciate the reviewer’s interest in this issue. Several non-mutually exclusive factors could account for the differences in OL morphology in vitro versus in vivo caused by Zdhhc9 loss. First, morphology in vivo may well be influenced by the axons and/or other extrinsic components around each OL that are not present in our primary cultures. Second, OL growth in vivo is highly 3-dimensional, whereas growth in culture is largely 2-dimensional – it may be difficult to support formation of spheroids (by definition, a 3-dimensional structure) in the latter situation. Finally, Zdhhc9 is absent in vivo from the beginning of development until the time points examined, whereas in our cultured OL experiments, Zdhhc9 shRNA is virally delivered to OPC cultures at DIV2 and likely acutely affects Zdhhc9 expression predominantly in committed OLs (following the switch to differentiation medium at DIV3). These differences may also affect the ability of other PATs or, potentially, palmitoylation-independent subcellular processes, to compensate for Zdhhc9 loss. We have more fully explained these points in our revised Discussion.

**Reviewer #2 (Public Review):**
This study provides an in-depth exploration of the impact of X-linked ZDHHC9 gene mutations on cognitive deficits and epilepsy, with a particular focus on the expression and function of ZDHHC9 in myelin-forming oligodendrocytes (OLs). These findings offer crucial insights into understanding ZDHHC9-related X-linked intellectual disability (XLID) and shed light on the regulatory mechanisms of palmitoylation in myelination. The experimental design and analysis of results are convincing, providing a valuable reference for further research in this field. However, upon careful review, I believe the article still needs further improvement and supplementation in the following aspects:(1) Regarding the subcellular localization experiment of ZDHHC9 mutants in OL, it is currently limited to in vitro cultured OL, lacking validation in vivo OL or myelin sheath. Additionally, it is necessary to investigate whether the abnormal subcellular localization of ZDHHC9 mutants affects their enzyme activity and palmitoylation modification of substrate proteins.

This is an important point but is technically challenging to address in vivo as it would likely require delivery of AAV to express ZDHHC9wt and XLID mutants specifically in OLs, preferably in the absence of endogenous ZDHHC9. We hope the reviewers would agree that this experiment is beyond the scope of the current study. However, we did compare the ability of ZDHHC9wt and XLID mutants to palmitoylate MBP, and to autopalmitoylate (sometimes used as a surrogate measure of PAT activity) in transfected heterologous cells. Although we recognize that this over-expression system is less physiological than a native OL, it has the benefit of being able to readily compare transfected wt vs mutant forms of ZDHHC9 with minimal contribution from endogenous ZDHHC9. Intriguingly, using this system, we found that autopalmitoylation activity of the XLID ZDHHC9-P150S mutant does not differ significantly from that of ZDHHC9wt, and that this mutant is still capable of palmitoylating MBP. Moreover, the R96W mutant, while impaired in autopalmitoylation, still palmitoylated MBP approximately 50% as effectively as ZDHHC9wt in our cell-based assay. These findings suggest that ZDHHC9-P150S and, probably, ZDHHC9-R96W mutants might still be able to palmitoylate substrates in OLs if they were properly localized. This possibility in turn suggests that impaired subcellular targeting in addition to, or instead of, impaired catalytic activity, may be a key factor in certain cases of ZDHHC9-associated XLID. We have expanded our Figure 8 (new panels 8E-G) to show these additional experiments and have summarized the conclusions above in our revised Discussion. We thank the reviewer for suggesting that we further investigate this issue.

(2) The experimental period (P21+21 days) using genetic labeling to track the development of myelinating cells may not be long enough. It is recommended to extend the observation time and analyze at more time points to more comprehensively reflect the impact of Zdhhc9 KO.

We appreciate this point from the reviewer but, regrettably, we did not maintain the PdgfraCreER; R26-EGFP; Zdhhc9 KO mouse line and hope the reviewer appreciates that it would take considerable time and effort to rederive this line and then perform the suggested extended time course experiments. However, we note for the reviewer that our preliminary studies did not reveal any effect of Zdhhc9 KO on the number of MOBP-EGFP+ OLs in 6-month-old mice (not shown), consistent with a model in which Zdhhc9 loss does not affect OPC-OL commitment per se.

(3) The author speculates that Zdhhc9 may regulate myelination by affecting the membrane localization of specific myelin proteins, but lacks direct experimental evidence to support this. It is suggested to detect the expression and distribution of relevant proteins in the myelin of Zdhhc9 KO mice.

We share the reviewer’s interest in this point but realized that it is more technically challenging to address than might be initially thought. The main protein we would implicate and seek to test is MBP, but we already found that there is no gross change in MBP distribution in vivo in Zdhhc9 KO mice (Fig 3A). However, an anti-MBP antibody recognizes all forms of MBP, not just the specific splice variants whose palmitoylation is affected by ZDHHC9 loss. Specifically assessing nanoscale distribution of these splice variants would require a way (e.g. anti-MBP splice form-specific antibodies that are compatible with immuno-EM) to distinguish these variants from other, non-palmitoylated forms of MBP. Although such an antibody could be an important tool, we hope the reviewers would agree that developing and characterizing such a reagent is beyond the scope of the current study.

We do, however, note that the lack of gross change in MBP distribution and levels in Zdhhc9 KO mice is consistent with the relatively mild phenotype of these mice, compared with shiverer (shi/shi) mice, in which MBP is completely lost. In shiverer, CNS compact myelin is almost absent (PMID: 671037; PMID: 88695; PMID: 460693) and, as the name suggests, mice display a shivering gait, and exhibit seizures and early death. In contrast, Zdhhc9 mice show only subtle behavioral deficits (PMID: 29944857). These differences are all consistent with a model in which Zdhhc9 KO mice, despite their significantly reduced MBP palmitoylation (Fig 8) have grossly normal distribution and levels of MBP when all splice variants are assessed (Fig 3, Fig 8). It is not inconceivable that Zdhhc9 KO mice have a nanoscale change in the distribution of MBP, particularly of specific palmitoylated splice variants, within myelin that profoundly affects myelin ultrastructure, without grossly altering MBP distribution. However, an alternative and not mutually exclusive possibility is that aberrant palmitoylation of other Zdhhc9 substrates accounts for, or contributes to, the abnormalities in myelin at the ultrastructural level. Addressing this issue would require a multi-pronged approach, not just to assess palmitoylation and distribution of such proteins in Zdhhc9 KO, but also to test whether they are direct Zdhhc9 substrates, in order to rule out indirect effects. We hope reviewers would agree that this is best left to a separate study. However, in our revised Discussion we now summarize what can be inferred regarding Zdhhc9-dependent effects on total and splicevariant specific distribution and levels of MBP.

(4) Although the article mentions the association of Zdhhc9 with intellectual disabilities, it does not involve behavioral analysis of Zdhhc9 KO mice. It is recommended to supplement some behavioral experimental data to support the important role of Zdhhc9 in maintaining normal cognitive function, enhancing the clinical relevance of the article.

We appreciate this point from the reviewer. The behavior of the same ZDHHC9 KO mouse line that we used was reported in PMID: 31747610 and in PMID: 29944857. In the former study, Zdhhc9 KO mice were reported to display seizures reminiscent of phenotypes in human patients with ZDHHC9 mutation. The latter study assessed performance of Zddhc9 KO mice in several tasks that test cognitive function. Specifically the KO mice were reported to display “altered behaviour in the open-field test, elevated plus maze and acoustic startle test that is consistent with a reduced anxiety level; a reduced hang time in the hanging wire test that suggests underlying hypotonia but which may also be linked to reduced anxiety [and] deficits in the Morris water maze test of hippocampal-dependent spatial learning and memory.”. We have incorporate these findings in our revised Discussion, where we summarize how these phenotypes are common, not just to human patients with ZDHHC9 mutation, but also to other human neurodevelopmental conditions and mouse models in which ID is a common feature.

(5) For the abnormal myelination observed in Zdhhc9 KO mice, including unmyelinated large-diameter axons and excessively myelinated small-diameter axons, the article lacks indepth research and explanation on the exact mechanism and mode of action of ZDHHC9 in regulating myelination.

We share the reviewer’s interest in this point but again note that gaining definitive insights into this issue is far from trivial. Convincing evidence of a causative mechanism would require an exhaustive identification of ZDHHC9 in vivo substrates, followed by point mutation of substrate palmitoylation site(s) to determine the extent to which palmitoylation of such protein(s) phenocopies ZDHHC9 loss. Nonetheless, it is possible to break this question down and to summarize what we do and do not know. For example, our experiments in cultured OLs show that ZDHHC9 loss causes call-autonomous deficits in morphological maturation of these cells. We also know that ZDHHC9 loss results in impaired palmitoylation of MBP, a direct substrate for ZDHHC9. Moreover, loss of ZDHHC9 at Golgi outposts in OLs (a phenotype observed with several XLID-associated mutant forms of ZDHHC9, even those with no significant loss of catalytic activity) correlates with intellectual disability. Together, these findings are consistent with a model in which ZDHHC9 action at OL Golgi outposts is critical for normal myelination. However, it is yet to be determined whether the key substrates of ZDHHC9 include MBP, other palmitoyl-proteins that are key constituents of CNS myelin, or proteins whose palmitoylation is important for myelin protein trafficking and targeting. Another non-mutually exclusive possibility is that ZDHHC9 acts at Golgi outposts but indirectly, for example to drive the expression of myelin protein genes. Future experiments, including but not limited to palmitoyl-proteomics in ZDHHC9 (OL-specific) KO mice, will be needed to provide more definitive insights into this issue. We have expanded our Discussion of links between ZDHHC9 mutation and impaired myelination to summarize the above points.

(6) The function of ZDHHC9 in OL may be related to the Golgi apparatus, but its exact role in these structures is still unclear. It is suggested to discuss in more detail the role of ZDHHC9 in the Golgi apparatus in the discussion section.

We appreciate this point, which we considered as related to point (5) above. In our revised Discussion we highlight how ZDHHC9 action at Golgi outposts may involve direct palmitoylation of myelin proteins, palmitoylation of proteins that direct myelin proteins to the myelin membrane and/or activation of gene expression programs that serve to drive myelination. We further note that these possibilities are not mutually exclusive.

(7) More experimental support and in-depth research are needed on the detailed mechanism of how ZDHHC9 and Golga7 cooperatively regulate MBP palmitoylation, and how this decrease in palmitoylation level leads to myelination defects.

This is another important point – our new experiments suggest that, although some XLID mutations markedly affect ZDHHC9’s ability to palmitoylate MBP, others do not, yet all of the mutant forms fail to localize to Golgi outposts. These findings are consistent with a model in which the subcellular location at which ZDHHC9 palmitoylates MBP, and potentially other substrates, is critical for normal myelination. Interestingly, despite their marked differences in basal catalytic activity (as assessed by autopalmitoylation), wt and all XLID forms of ZDHHC9 appear to show enhanced activity (measured by both auto- and MBP palmitoylation) in the presence of ZDHHC9, suggesting that the association with Golga7 (which also localizes to Golgi outposts) is central to ZDHHC9 activity. This model is also highly consistent with the biased expression of Golga7 in OLs, compared to other CNS cell types (Fig 1E, 1F). Moreover, XLID-associated mutant forms of ZDHHC9 also show reduced protein stability and are impaired in their ability to form complexes with Golga7 (also known as Golgi Complex Protein 16kDa; GCP16; PMID: 37035671). Failure of ZDHHC9 XLID mutants to localize to Golgi outposts may thus be due to aberrant trafficking of mutant ZDHHC9 per se, but may also involve impaired association/stabilization of ZDHHC9/Golga7 complexes at these locations. Again, it is possible that either or both of these mechanisms, which are not mutually exclusive, contribute to impaired MBP palmitoylation and/or myelination deficits. We summarize these points in our revised Discussion.

In summary, it is recommended that the authors address the above issues through additional experiments and improved discussions to further strengthen the credibility and clinical relevance of the article.
**Recommendations for the authors:**

**Reviewer #1 (Recommendations For The Authors):**
No gross changes were detected in OL development in Zdhhc9 KO mice and analyses from P28 Zdhhc9 KO mice crossed with Mobp-EGFP reporter mice did not show changes in EGFP+ OL differentiation (Figure 3). However, and given the observed subcellular localization of ZDHHC9 in OL processes (Figure 2) and the observation that the percentage of unmyelinated axons is increased in Zdhhc9 KO (Figure 6), ***early time points to examine the differentiated pools of OLs and their capacity to extend processes/contact axons need to be considered***.

We appreciate this point, but due to the order in which experiments were performed, the ZDHHC9 KO mouse colony that we maintained after initial submission of this work contains homozygous MOBP-EGFP, but not the mT/mG transgene that would be most optimal for the proposed experiment. We hope the reviewer appreciates that it would take considerable time and effort regarding mouse breeding to cross out the MOBP and add back the mT/mG. We nonetheless appreciate the importance of the point raised and therefore examined an earlier developmental time point (P21, 3 weeks) to quantify OLs and NG2+ OPCs. In our updated Fig 3C1-C3, we use Mobp-EGFP mice to show that Zdhhc9 KO does not significantly affect the number of EGFP+ OLs at this time point in the cortex, corpus callosum and spinal cord. We also show that in corpus callosum, Zdhhc9 KO does not significantly affect the number of NG2+ OPCs at this earlier time point (Fig 3D, E). Furthermore, immunostaining to detect BCAS1, a marker of pre-mature OLs, also revealed no qualitative difference with ZDHHC9 loss at P21. We show representative images from these BCAS1 experiments in an updated Fig S3. While these new experiments do not address the morphology of OLs in Zdhhc9 KO, they do provide further evidence that deficits in myelination in young Zdhhc9 KO mice (Figure 6) are not likely due to gross differences in OPC or OL numbers during development.

The authors observed defects in Zdhhc9 KO OL protrusions that they attributed to abnormal OL membrane expansion (Fig 4 and 5). Can they show evidence for this?

This is an important point, and we appreciate the opportunity to explain the reasoning behind our initial statement more fully, while noting that other explanations are possible. Fig 5B (an Imaris-assisted reconstruction using the EGFP cell fill/morphology marker) highlights large spheroid-like distensions along OL processes. We reason that these spheroids are enclosed by the OL lipid membrane because if the membrane were ruptured, the EGFP signal would likely diffuse. This in turn suggests that the caliber of the OL process at the position of the spheroid is grossly abnormal i.e. the membrane has hyper-expanded. Given that OL membrane growth during myelination extends in two directions, i.e., spiral growth to the axonal surface and longitudinal growth along the axon, it is possible that spheroid-like structures are formed by uneven myelin growth. We recognize that we cannot yet conclude whether and how spheroid formation might be linked to the myelination deficit that we observe in Zdhhc9 KO mice.

However, defining the subcellular mechanism for spheroid formation may provide further insights into this issue. We have therefore largely retained the original statement but have added the reasoning above to our revised Discussion.

The authors report that Zdhhc9 KO primary and secondary branches in OL were longer, some contained spheroid-like swellings and the OL protrusion complexity was higher. However, these data is partially contradictory to what they show in OL differentiation experiments in vitro (Fig 7). There is also no evidence for increased membrane expansion in Zdhhc9 knockdown myelin forming cells in culture. How do they reconcile these different findings?

We appreciate the reviewer’s interest in this issue. Several non-mutually exclusive factors could account for the differences in OL morphology in vitro versus in vivo caused by Zdhhc9 loss. First, morphology in vivo may well be influenced by the axons and/or other extrinsic components around each OL that are not present in our primary cultures. Second, OL growth in vivo is highly 3-dimensional, whereas growth in culture is largely 2-dimensional – it may be difficult to support formation of spheroids (by definition, a 3-dimensional structure) in the latter situation. Finally, Zdhhc9 is absent in vivo from the beginning of development until the time points examined, whereas in our cultured OL experiments, Zdhhc9 shRNA is virally delivered to OPC cultures at DIV2 and likely acutely affects Zdhhc9 expression predominantly in committed OLs (following the switch to differentiation medium at DIV3). These differences may also affect the ability of other PATs or, potentially, palmitoylation-independent subcellular processes, to compensate for Zdhhc9 loss. We have more fully explained these points in our revised Discussion.

Page 7: "The OL processes in this culture condition correspond to large lipid-rich membranous sheets that form spiral membrane expansion on axons in vivo (49)." At which stage are authors referring to? OL processes are extended in culture before membrane formation and this is not clear here. In a 3-days differentiation culture, most OLs have not yet formed a myelin sheath (eg., Figure 2 in Zuchero et al., 2015, Dev Cell).

We appreciate the reviewer highlighting this point. We first note that our oligodendrocyte (OL) culture conditions differ from the immunopanning method used by Zuchero et al., 2015 (original reference (Emery and Dugas, 2013)), which may affect the time course and progression of OL process elaboration and/or myelin sheath formation. We further note that in our cultures most EGFP+ processes are also MBP+ at the time point examined (strictly 3 days plus 9 hours post-differentiation). It thus seems likely that these MBP+ structures largely correspond to the MBP+ wrapping sheaths that occur in vivo, so we have therefore retained our original statement but have added this further explanation.

Minor: Figure 6 (Legend): Time points should be indicated throughout the panels.

We have added this information as requested

**Reviewer 2 Recommendations for the Authors:**
(1) Regarding the subcellular localization experiment of ZDHHC9 mutants in OL, it is currently limited to in vitro cultured OL, lacking validation in vivo OL or myelin sheath. Additionally, it is necessary to investigate whether the abnormal subcellular localization of ZDHHC9 mutants affects their enzyme activity and palmitoylation modification of substrate proteins.

We thank the reviewer for raising this point. New data in our revised Figure 8 compares autopalmitoylation (sometimes used as a surrogate measure of PAT activity) of ZDHHC9wt and XLID mutants, and their ability to palmitoylate MBP in transfected cells. Intriguingly, we found that autopalmitoylation activity of the ZDHHC9-P150S mutant does not differ significantly from that of ZDHHC9wt, and that this mutant is still capable of palmitoylating MBP. Moreover, the R96W mutant, while impaired in autopalmitoylation, still palmitoylated MBP approximately 50% as effectively as ZDHHC9wt in our cell-based assay. These findings suggest that ZDHHC9-P150S and, probably, ZDHHC9-R96W mutants might still be able to palmitoylate substrates in OLs if they were properly localized. This possibility in turn suggests that impaired subcellular targeting in addition to, or instead of, impaired catalytic activity, may be a key factor in certain cases of ZDHHC9-associated XLID. We have expanded our Figure 8 to show these new experiments and have summarized the conclusions above in our revised Discussion. We thank the reviewer for suggesting that we further investigate this issue.

(2) The experimental period (P21+21 days) using genetic labeling to track the development of myelinating cells may not be long enough. It is recommended to extend the observation time and analyze at more time points to more comprehensively reflect the impact of Zdhhc9 KO.

We appreciate this point from the reviewer but, regrettably, we did not maintain the PdgfraCreER; R26-EGFP; Zdhhc9 KO mouse line and hope the reviewer appreciates that it would take considerable time and effort to rederive this line and then perform the suggested extended time course experiments. However, we note for the reviewer that our preliminary studies did not reveal any effect of Zdhhc9 KO on the number of MOBP-EGFP+ OLs in 6-month-old mice (not shown), consistent with a model in which Zdhhc9 loss does not affect OPC-OL commitment per se.

(3) The author speculates that Zdhhc9 may regulate myelination by affecting the membrane localization of specific myelin proteins, but lacks direct experimental evidence to support this. It is suggested to detect the expression and distribution of relevant proteins in the myelin of Zdhhc9 KO mice.

We share the reviewer’s interest in this point but realized that it is more technically challenging to address than might be initially thought. The main protein we would implicate and seek to test is MBP, but we already found that there is no gross change in MBP distribution in vivo in Zdhhc9 KO mice (Fig 3A). However, an anti-MBP antibody recognizes all forms of MBP, not just the specific splice variants whose palmitoylation is affected by ZDHHC9 loss. Specifically assessing nanoscale distribution of these splice variants would require a way (e.g. am anti-MBP splice form-specific antibody that is compatible with immuno-EM) to distinguish these variants from other, non-palmitoylated forms of MBP. Although such an antibody could be an important tool we hope the reviewers would agree that developing and characterizing such a reagent is beyond the scope of the current study.

We do, however, note that the lack of gross change in MBP distribution and levels in Zdhhc9 KO mice is consistent with the relatively mild phenotype of these mice, compared with shiverer (shi/shi) mice, in which MBP is completely lost. In shiverer, CNS compact myelin is almost absent (PMID: 671037; PMID: 88695; PMID: 460693) and, as the name suggests, mice display a shivering gait, and exhibit seizures and early death. In contrast, Zdhhc9 mice show only subtle behavioral deficits (PMID: 29944857). These differences are all consistent with a model in which Zdhhc9 KO mice, despite their significantly reduced MBP palmitoylation (Fig 8) have grossly normal distribution and levels of MBP when all splice variants are assessed (Fig 3, Fig 8). It is not inconceivable that Zdhhc9 KO mice have a nanoscale change in the distribution of MBP, particularly of specific palmitoylated splice variants, within myelin that profoundly affects myelin ultrastructure, without grossly altering MBP distribution. However, an alternative and not mutually exclusive possibility is that aberrant palmitoylation of other

Zdhhc9 substrates accounts for, or contributes to, the abnormalities in myelin at the ultrastructural level. Addressing this issue would require a multi-pronged approach, not just to assess palmitoylation and distribution of such proteins in Zdhhc9 KO, but also to test whether they are direct Zdhhc9 substrates, in order to rule out indirect effects. We hope reviewers would agree that this is best left to a separate study. However, in our revised Discussion we now summarize what can be inferred regarding Zdhhc9-dependent effects on total and splicevariant specific distribution and levels of MBP.

(4) Although the article mentions the association of Zdhhc9 with intellectual disabilities, it does not involve behavioral analysis of Zdhhc9 KO mice. It is recommended to supplement some behavioral experimental data to support the important role of Zdhhc9 in maintaining normal cognitive function, enhancing the clinical relevance of the article.

We appreciate this point from the reviewer. The behavior of the same ZDHHC9 KO mouse line that we used was reported in PMID: 31747610 and in PMID: 29944857. In the former study, Zdhhc9 KO mice were reported to display seizures reminiscent of phenotypes in human patients with ZDHHC9 mutation. The latter study assessed performance of Zddhc9 KO mice in several tasks that test cognitive function. Specifically the KO mice were reported to display “altered behaviour in the open-field test, elevated plus maze and acoustic startle test that is consistent with a reduced anxiety level; a reduced hang time in the hanging wire test that suggests underlying hypotonia but which may also be linked to reduced anxiety [and] deficits in the Morris water maze test of hippocampal-dependent spatial learning and memory.”. We have incorporate these findings in our revised Discussion, where we summarize how these phenotypes are common, not just to human patients with ZDHHC9 mutation, but also to other human neurodevelopmental conditions and mouse models in which ID is a common feature.

(5) For the abnormal myelination observed in Zdhhc9 KO mice, including unmyelinated large-diameter axons and excessively myelinated small-diameter axons, the article lacks indepth research and explanation on the exact mechanism and mode of action of ZDHHC9 in regulating myelination.

We share the reviewer’s interest in this point but again note that gaining definitive insights into this issue is far from trivial. Convincing evidence of a causative mechanism would require an exhaustive identification of ZDHHC9 in vivo substrates, followed by point mutation of substrate palmitoylation site(s) to determine the extent to which palmitoylation of such protein(s) phenocopies ZDHHC9 loss. Nonetheless, it is possible to break this question down and to summarize what we do and do not know. For example, our experiments in cultured OLs show that ZDHHC9 loss causes call-autonomous deficits in morphological maturation of these cells. We also know that ZDHHC9 loss results in impaired palmitoylation of MBP, a direct substrate for ZDHHC9. Moreover, loss of ZDHHC9 at Golgi outposts in OLs (a phenotype observed with several XLID-associated mutant forms of ZDHHC9, even those with no significant loss of catalytic activity) correlates with intellectual disability. Together, these findings are consistent with a model in which ZDHHC9 action at OL Golgi outposts is critical for normal myelination. However, it is yet to be determined whether the key substrates of ZDHHC9 include MBP, other palmitoyl-proteins that are key constituents of CNS myelin, or proteins whose palmitoylation is important for myelin protein trafficking and targeting. Another non-mutually exclusive possibility is that ZDHHC9 acts at Golgi outposts but indirectly, for example to drive the expression of myelin protein genes. Future experiments, including but not limited to palmitoyl-proteomics in ZDHHC9 (OL-specific) KO mice, will be needed to provide more definitive insights into this issue. We have expanded our Discussion of links between ZDHHC9 mutation and impaired myelination to summarize the above points.

(6) The function of ZDHHC9 in OL may be related to the Golgi apparatus, but its exact role in these structures is still unclear. It is suggested to discuss in more detail the role of ZDHHC9 in the Golgi apparatus in the discussion section.

We appreciate this point, which we considered as related to point (5) above. In our revised Discussion we highlight how ZDHHC9 action at Golgi outposts may involve direct palmitoylation of myelin proteins, palmitoylation of proteins that direct myelin proteins to the myelin membrane and/or activation of gene expression programs that serve to drive myelination. We further note that these possibilities are not mutually exclusive.

(7) More experimental support and in-depth research are needed on the detailed mechanism of how ZDHHC9 and Golga7 cooperatively regulate MBP palmitoylation, and how this decrease in palmitoylation level leads to myelination defects.

This is another important point – our new experiments suggest that, although some XLID mutations markedly affect ZDHHC9’s ability to palmitoylate MBP, others do not, yet all of the mutant forms fail to localize to Golgi outposts. These findings are consistent with a model in which the subcellular location at which ZDHHC9 palmitoylates MBP, and potentially other substrates, is critical for normal myelination. Interestingly, despite their marked differences in basal catalytic activity (as assessed by autopalmitoylation), wt and all XLID forms of ZDHHC9 appear to show enhanced activity (measured by both auto- and MBP palmitoylation) in the presence of ZDHHC9, suggesting that the association with Golga7 (which also localizes to Golgi outposts) is central to ZDHHC9 activity. This model is also highly consistent with the biased expression of Golga7 in OLs, compared to other CNS cell types (Fig 1E, 1F). Moreover, XLID-associated mutant forms of ZDHHC9 also show reduced protein stability and are impaired in their ability to form complexes with Golga7 (also known as Golgi Complex Protein 16kDa; GCP16; PMID: 37035671). Failure of ZDHHC9 XLID mutants to localize to Golgi outposts may thus be due to aberrant trafficking of mutant ZDHHC9 per se, but may also involve impaired association/stabilization of ZDHHC9/Golga7 complexes at these locations. Again, it is possible that either or both of these mechanisms, which are not mutually exclusive, contribute to impaired MBP palmitoylation and/or myelination deficits. We summarize these points in our revised Discussion.